# Perirhinal firing patterns are sustained across large spatial segments of the task environment

Jeroen J. Bos[1,2,*], Martin Vinck[1,3,*], Laura A. van Mourik-Donga[1,2], Jadin C. Jackson[1,4], Menno P. Witter[5] & Cyriel M.A. Pennartz[1,2]

Spatial navigation and memory depend on the neural coding of an organism's location. Fine-grained coding of location is thought to depend on the hippocampus. Likewise, animals benefit from knowledge parsing their environment into larger spatial segments, which are relevant for task performance. Here we investigate how such knowledge may be coded, and whether this occurs in structures in the temporal lobe, supplying cortical inputs to the hippocampus. We found that neurons in the perirhinal cortex of rats generate sustained firing patterns that discriminate large segments of the task environment. This contrasted to transient firing in hippocampus and sensory neocortex. These spatially extended patterns were not explained by task variables or temporally discrete sensory stimuli. Previously it has been suggested that the perirhinal cortex is part of a pathway processing object, but not spatial information. Our results indicate a greater complexity of neural coding than captured by this dichotomy.

[1] Swammerdam Institute for Life Sciences, Center for Neuroscience, Faculty of Science, University of Amsterdam, 1098 XH Amsterdam, The Netherlands. [2] Research Priority Program Brain and Cognition, University of Amsterdam, 1098 XH Amsterdam, The Netherlands. [3] Ernst Strüngmann Institute for Neuroscience in Cooperation with Max Planck Society, Deutschordenstraße 46, 60528 Frankfurt, Germany. [4] Medtronic, 7000 Central Avenue NE, Minneapolis, Minnesota 55432, USA. [5] Kavli Institute for Systems Neuroscience, Centre for Neural Computation, Norwegian University of Science and Technology, DMF, NTNU PO Box 8905, NO-7491 Trondheim, Norway. * These authors contributed equally to this work. Correspondence and requests for materials should be addressed to C.M.A.P. (email: c.m.a.pennartz@uva.nl).

Over the past decades, research on neural mechanisms underlying spatial navigation and memory has largely focused on the neural coding of an animal's local position, heading direction and velocity of movement[1–5]. While these parameters are needed to mediate detailed navigation, it is equally useful for animals to encode macrogeometrical knowledge to navigate in large-scale environments to reach a distant goal. The ability to orient and navigate successfully in large-scale environments is also referred to as 'topographical orientation'[6] and the lack of this capacity is one of the hallmarks of Alzheimer's disease[6–8]. Topographical orientation can be illustrated by a car driver who navigates through a city to get from district A to a house in a remote quarter B. Given this task, she first needs to apply large-scale knowledge of, for example, T-junctions and neighbourhoods to map an overall route for navigation. Only when getting close to B does she require more detailed knowledge, for example, of which houses she will pass in succession. Navigation through environments may be facilitated by ventral hippocampal and entorhinal cell populations, showing larger scales of spatial coding than their dorsal counterparts[1,9]. However, both dorsal and ventral hippocampal cells have place fields scattered across the environment, and it remains unknown whether and how chunks of environments or spatial trajectories, as demarcated by decision points, are coded.

The hippocampus is a key structure for coding an animal's detailed location in space[2–5,10] and time[11,12], and for linking significant behavioural events to this spatiotemporal framework[13–16]. Two major open questions are how afferent structures in the medial temporal lobe supply the hippocampus with information to build these complex representations, and whether they harbour neurons coding other large-scale knowledge of an animal's task environment than found in ventral hippocampal–entorhinal circuits. A dominant hypothesis states that there exist two segregated routes that supply what and where information to the hippocampus, centred on the perirhinal (PRH)–lateral entorhinal cortices and the postrhinal–medial entorhinal cortices, respectively[15,17–20]. This hypothesis is supported, amongst others, by evidence for spatial representations encoded by grid cells in medial entorhinal cortex[1,5] and for PRH–lateral entorhinal cortex functions in object discrimination and recognition[18,19,21–24].

An alternative view of PRH holds that it plays a more general role than merely coding representations of discrete, single objects. According to this more general account, PRH mediates discrimination and learning of complex configurations in the environment, which may include multiple items and environmental context[25–28]. Such a unitizing function of the PRH, serving to integrate two or more previously separate items into a single representation[29], is supported by lesion studies[23,30,31], but it remains unclear whether, and how, PRH encodes neural representations of environmental context. This also raises the question of whether PRH function extends from single objects to larger units of information processed within a task context, representing chunks of the environment. Spatially restricted firing of PRH cells has been observed before, but was not stable across different task conditions[32]. Here we tested the unitizing hypothesis by recording PRH cells in a sensory discrimination task set on a figure-8 maze, and demonstrate that PRH cells do perform integrative operations, globally specifying where, in the task context, an animal is located. Simultaneously recorded dorsal hippocampal CA1 and sensory neocortical cells do not exhibit this integrative property.

## Results

Rats navigated a figure-8 maze while performing a visual discrimination task (Fig. 1a,b). This task combined the presentation of discrete items with spatial choice behaviour. On each trial, a visual target stimulus (CS + ) and distracter (CS − ) were shown on two screens flanking the initial segments of each side arm. To obtain rewards, rats had to pass a decision (bifurcation) point, choosing the arm that was flanked by the CS + , whereas following the CS − did not produce reward. After visiting a reward site, rats returned to the middle lane, where they could initiate a new trial by breaking a photobeam after a sound cue was given at the end of an intertrial interval (ITI). Additional tactile cues were provided by way of sandpaper-covered initial segments of the side arms. While rats were travelling across these segments, they were allowed to change their initial decision until passing a 'point of no return' (PNR) situated at the end of the sandpaper-covered segments. This passage was recorded by a photobeam break, and similarly we recorded the moment when rats passed the point of returning to the middle lane (point of return to middle, PRM). Overall, this task presents a rich combination of spatial and sensory variables as well as elements critical to behavioural decision-making, thereby offering to gain insight in PRH coding of a multitude of behaviourally relevant parameters. Using a 144-channel quad-drive (Supplementary Fig. 1; $N = 3$ rats), we made ensemble recordings from four brain areas simultaneously. Here we will primarily focus on firing patterns in PRH (518 cells) and compare these to recordings from hippocampal area CA1 (660 cells), somatosensory cortex (S1 barrel field, S1BF; 458 cells) and visual cortex (V1M; 43 cells, presented in Supplementary Fig. 4).

**Sustained PRH firing patterns**. In PRH we identified several cell types according to the item presented and task phase in which their firing activity exhibited significant changes with respect to baseline (Supplementary Fig. 2). Cells frequently responded to task elements such as the auditory and visual cues, and these firing responses were roughly similar to neural correlates described earlier for rodent and monkey PRH cells in an object-cued choice task[33] and temporal-order memory task[34], respectively. In addition, however, we identified a large fraction of PRH cells showing a sustained change in firing rate in the left or right arm after the animal made an initial choice. Below we will first present analyses of PRH firing behaviour in the time domain, because our behavioural set-up allowed us to accurately correlate firing-rate changes to the passage of the PNR and PRM in time. Moreover, the repetitive nature of the task permitted insight in the consistency of firing correlates over trials. Subsequently, analyses in the spatial domain will be presented.

Whereas firing patterns in area CA1 and S1BF exhibited brief, transient peaks in firing rate along the arms, PRH neurons often showed a prolonged firing-rate change, usually stretching from the spatial decision point, just before reaching the PNR, to the PRM. These differences were already visible by comparing single trials into the left or right arm (Fig. 1c,d). A significant left–right discrimination was made by 72.2% of the total number of PRH neurons (Methods and Supplementary Table 1); PRH neurons recorded on the same tetrode were not correlated in their left–right selectivity. These discriminatory patterns were marked by an increment in firing rate on one side but not the other (Figs 1e and 2a), by a decrement on one side (Fig. 1e), or bidirectional left–right modulations (Fig. 2b and Supplementary Table 2; firing-rate changes are denoted as increments and decrements relative to the baseline, computed during the ITI, spent in the middle lane). A minority of units showed a sustained increase or decrease in both arms, thus distinguishing the middle lane from both side loops (Supplementary Fig. 2e,f). In contrast to PRH, CA1 (Figs 1f and 2c) and S1BF neurons (Fig. 2d) exhibited transient changes in firing activity. For area CA1, showing

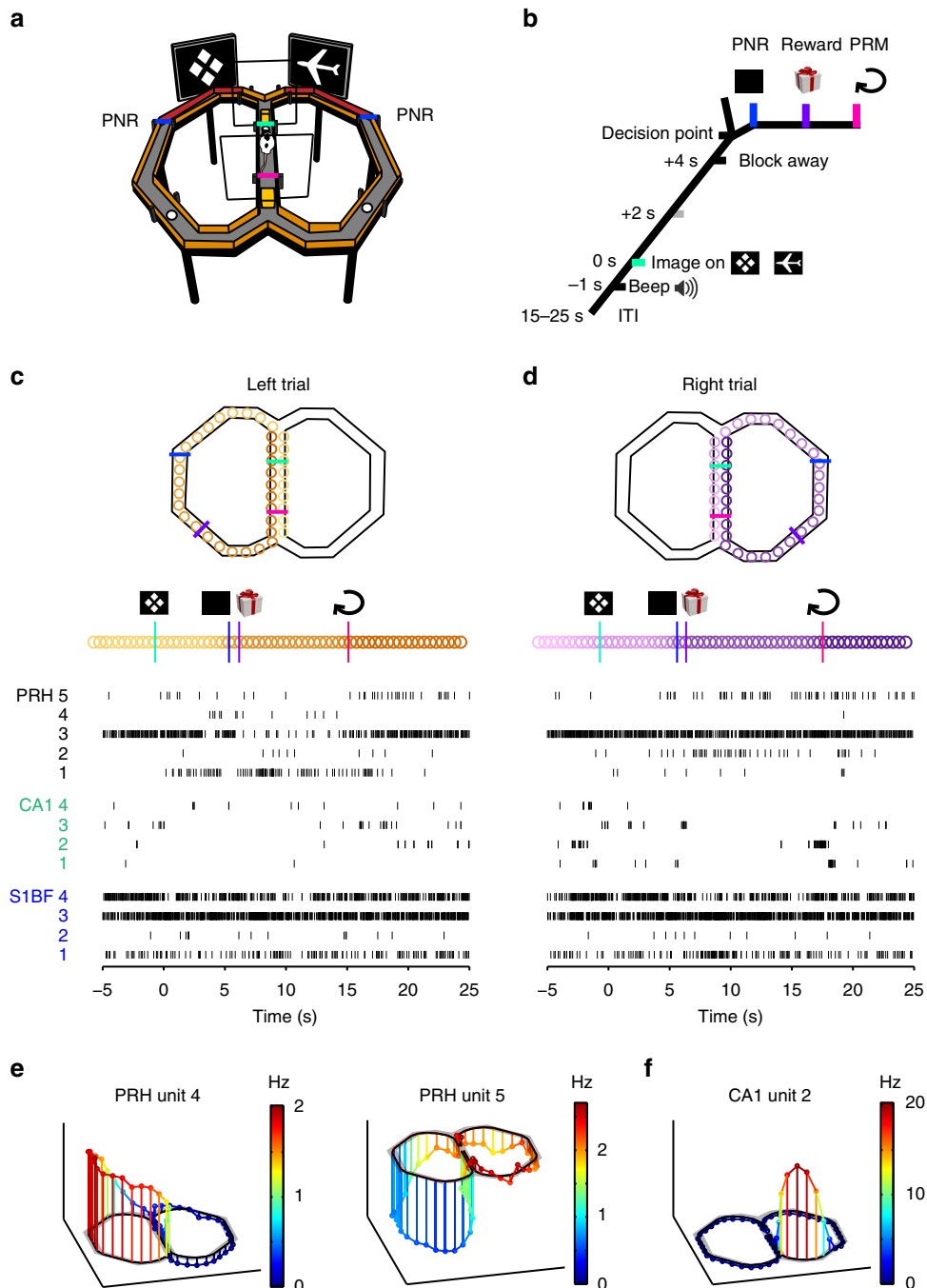

**Figure 1 | Single trial examples of firing patterns in PRH contrasted to simultaneously recorded units from hippocampus and barrel cortex. (a,b)** Spatial and temporal layout of the task. During the ITI rats were waiting between two transparent barriers (rectangles). Speaker symbolizes sound cue, 1s before image onset. Reward sites in side arms are indicated by white disks. Initial segments of both arms contain sandpaper (red) until PNR (blue bar). **(c,d)** During single, correct left and right trials, a subset of PRH cells (black; 1–5) showed sustained firing-rate increments or decrements during locomotion selectively in one of the side arms. For instance, PRH cell 3 shows a high tonic firing rate except for a sustained decrement selectively in the left arm. Coloured circles represent successive positions (orange: left trial with lightest colours in middle lane representing period before onset of visual cue; dark orange represents return to middle lane; purple: idem but for right trial). In contrast, hippocampal (green; 1–4) and S1BF cells (blue; 1–4) showed transient or no changes when travelling across the left or right arm. For instance, although S1BF cells 3 and 4 were tonically active, they showed a sustained decrement neither in the left nor right arm. **(e)** Two PRH cells (PRH unit 4 and 5; same cells as in **c**) showed left–right selectivity in firing rate (colour-coded) along the maze trajectories. **(f)** Same as (**e**), but now showing a hippocampal place field (CA1 unit 2; same cell as in **d**).

left–right discrimination in 61.2% of the total number of cells, selectivity could be ascribed to place fields located on the left or right arm. In S1BF, a similar pattern of 'punctate' responses was found in 39.7% of the total number of left–right selective cells (which however should not be considered place cells).

Below we will primarily focus on the sustained nature of PRH responses.

The firing-rate contrast between the two side arms was expressed in a normalized firing-rate score quantifying the difference between a cell's preferred minus non-preferred arm,

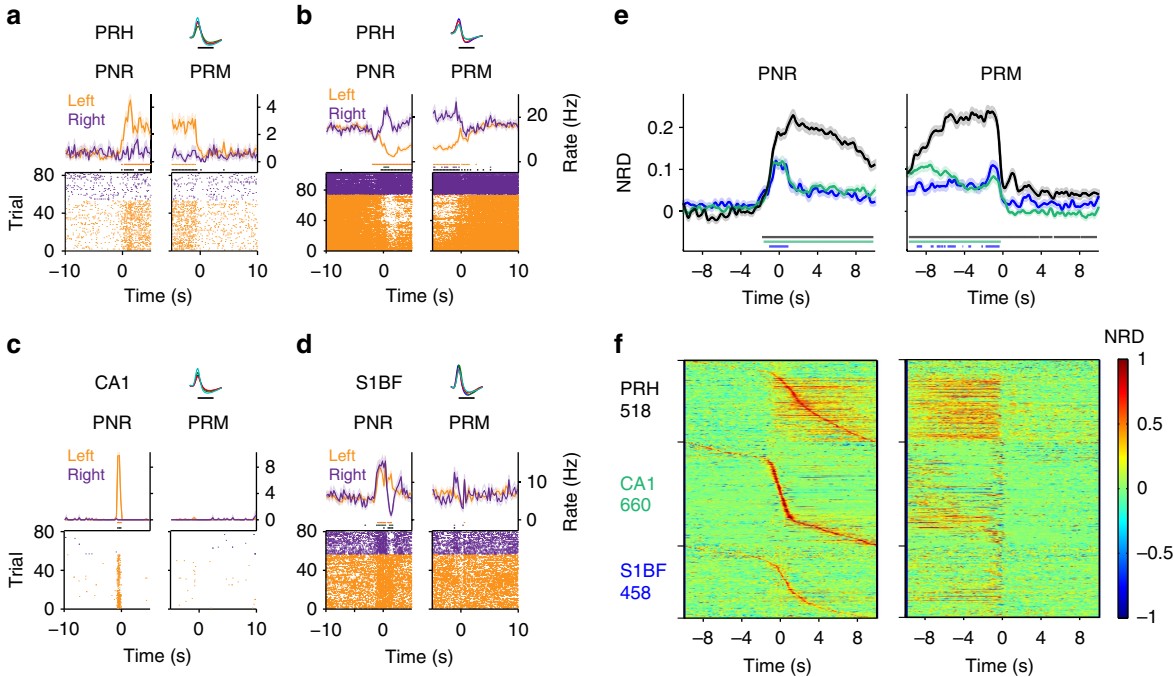

**Figure 2 | Sustained firing patterns occur selectively in PRH cells.** (**a–f**) Firing-rate histograms (mean ± s.e.m.) are synchronized on passing the PNR (left panels) and the PRM (right). Horizontal lines below the two curves of each graph indicate statistical significance of left (orange) and right (purple) trials relative to baseline (Wilcoxon signed-rank test, $\alpha = 0.05$; see Methods), or of the left–right difference (black, Mann–Whitney $U$-test with false discovery rate correction, $\alpha = 0.05$). (**a**) PRH cell increases firing rate throughout left but not right arm. Spike waveforms are shown on upper right (scale bar, 0.5 ms). (**b**) Another PRH neuron is bidirectionally modulated (left arm: decrement; right arm: increment). (**c,d**) CA1 (**c**) and S1BF (**d**) neurons fire transiently due to place field in left arm (**c**) and around PNR and return to middle lane (**d**). (**e**) Average NRD (preferred–non-preferred arm) for PRH (black), CA1 (green) and S1BF (blue). Horizontal lines below the two curves of each graph indicate statistical significance (Wilcoxon signed-rank test, $\alpha = 0.01$) of rate differences with colours corresponding to the three brain areas. (**a–e**) Each test bin needed to meet the $\alpha$ threshold against 2 s of contiguous baseline bins to be designated as significant (Methods). (**f**) Population overview of firing-rate differences for all three areas (cells ordered by moment of peak firing rate; cell counts on left).

normalized rate difference, NRD (averaged across cells; Fig. 2e; the arm correlated to the highest mean firing rate of a cell was labelled the 'preferred' arm of that cell). This analysis showed that the mean level of discrimination in CA1 and S1BF was much weaker than in PRH. Discriminatory firing by PRH neurons strongly increased around the passage of the PNR and steeply decreased when passing the PRM. In addition to this steep decrease, a residual significant rate difference remained present when the rat had returned to the middle lane, an effect that was not observed in area CA1 or S1BF (Fig. 2e). The strong left–right contrast of PRH cells was further confirmed by considering an unbiased, shuffle-corrected statistical measure of left–right discrimination, the discrimination score. All three recorded regions harboured a considerable fraction of cells reaching a significant discrimination score (PRH: 35.8%; CA1: 23.0%; and S1BF: 10.2% of the total number of cells per area), but the PRH cell population stood out by the discrimination score being elevated for a longer duration in between the PNR and PRM than in the other three areas (Supplementary Fig. 3). A population overview of all recorded cells showed the widespread occurrence of sustained left–right differences in PRH relative to CA1 and S1BF (Fig. 2f); the same pattern was observed for individual rats (Supplementary Table 1). This overview clarifies that the modest, but sustained mean between-arm differences in CA1 and S1BF (Fig. 2e) arise from the consecutive line-up of individual, transient responses occurring on one of the two arms (Fig. 2f). Thus, interregional comparisons indicate that all three recorded areas contain cells demarcating the engagement in left versus right choice trials,

whereas only PRH neurons show sustained discriminatory firing throughout this engagement.

**Sustained firing patterns in the spatial domain.** The temporally sustained nature of discriminatory PRH firing prompted us to investigate more systematically how it transfers to the spatial domain. When firing rates were plotted as a function of the rat's spatial position on the maze, the sustained firing-rate changes in PRH were indeed observed across the left and right arms, most prominently between the two branch points on the maze (that is, the decision point and return branch point, which the animals reached just before passing the PRM photobeam; Figs 1e and 3). Observing that the left–right difference scores plotted as a function of linearized position showed widespread discrimination across the PRH population (Fig. 3c,d), we next asked whether this discrimination in the spatial domain is attributable to a systematic contrast between firing-rate increments in one arm versus decrements in the other arm. In addition to extended firing-rate increments expressed on the cell's preferred arm (Fig. 3e), a sustained decrement in the non-preferred arm clearly contributed to the discrimination (Fig. 3f; see also Figs 2b and 3b). In contrast, such decrements were not observed in CA1 (Fig. 3g–i) or neurons in sensory cortices (Supplementary Fig. 4). Moreover, the consistency of left–right differences across spatial bins was higher in PRH than other structures (Fig. 3j,k and Supplementary Fig. 4). The joint expression of sustained increments and decrements by the same cells suggests that PRH firing patterns contain an activating and a deactivating

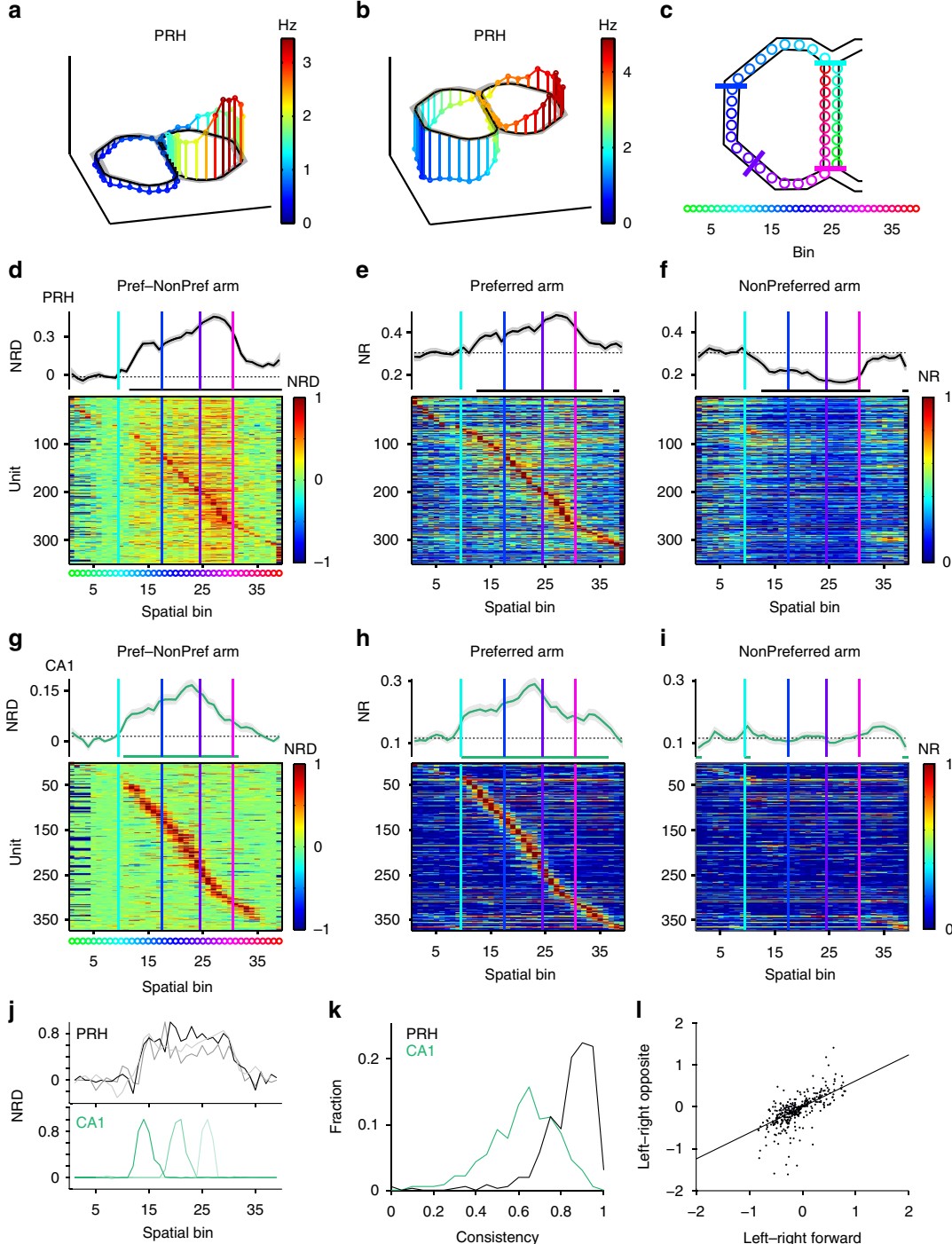

**Figure 3 | PRH firing studied as function of place reveals integration within task space.** (**a,b**) Two PRH cells showing left–right selectivity in firing rate (colour-coded; different units than Fig. 1). (**c,d**) NRD in PRH as a function of linearized maze position. Colour code below (**d**) corresponds to maze positions in **c**. Only units ($N = 349$) with a discrimination score $> 0.5$ were included in this graph. The mean NRD is plotted on top of the colour-coded survey (baseline: dashed line; significance with respect to baseline: horizontal line). Vertical coloured lines correspond to the spatial decision point (cyan), PNR (blue), reward site (purple) and return to middle (pink; see also **c**). (**e,f**) The NRD (**d**) can be decomposed into sustained PRH firing increments in the preferred arm (Pref) (**e**), and prolonged decrements in the non-preferred (NonPref) arm (**f**). NR is normalized firing rate. Mean NR is plotted on top of the colour-coded surveys. (**g–i**) Same for hippocampal CA1 neurons ($N = 373$), showing transient peak responses in one arm, as expected for place fields. For data on sensory neocortex, see Supplementary Fig. 4. The mean NRD in **g** shows spatially extended enhancement due to the fact that CA1 cells with place fields in the preferred arm contribute punctate responses, which add up in the mean NRD. No such enhancement is seen in the CA1 non-preferred arm plot (**i**) because, by definition, this arm is devoid of strong place fields. (**j**) Three examples of PRH (top, black) and hippocampal (bottom, green) neurons contrasting sustained versus transient NRDs as a function of linearized maze position. (**k**) Frequency histogram of the consistency of left–right discriminatory firing across spatial bins for PRH (black) and CA1 (green). High consistency indicates that the firing-rate differences between the preferred and non-preferred sides are uniformly maintained across the spatial bins of the arms. The consistency of PRH was higher than the consistency of CA1 (Mann–Whitney $U$-test, $Z = 17.76$, $P = 0$). (**l**) Left–right firing-rate differences were correlated between locomotion in forward and opposite heading directions. Pearson's correlation coefficient $r = 0.62$ ($P = 1.56 \times 10^{-38}$).

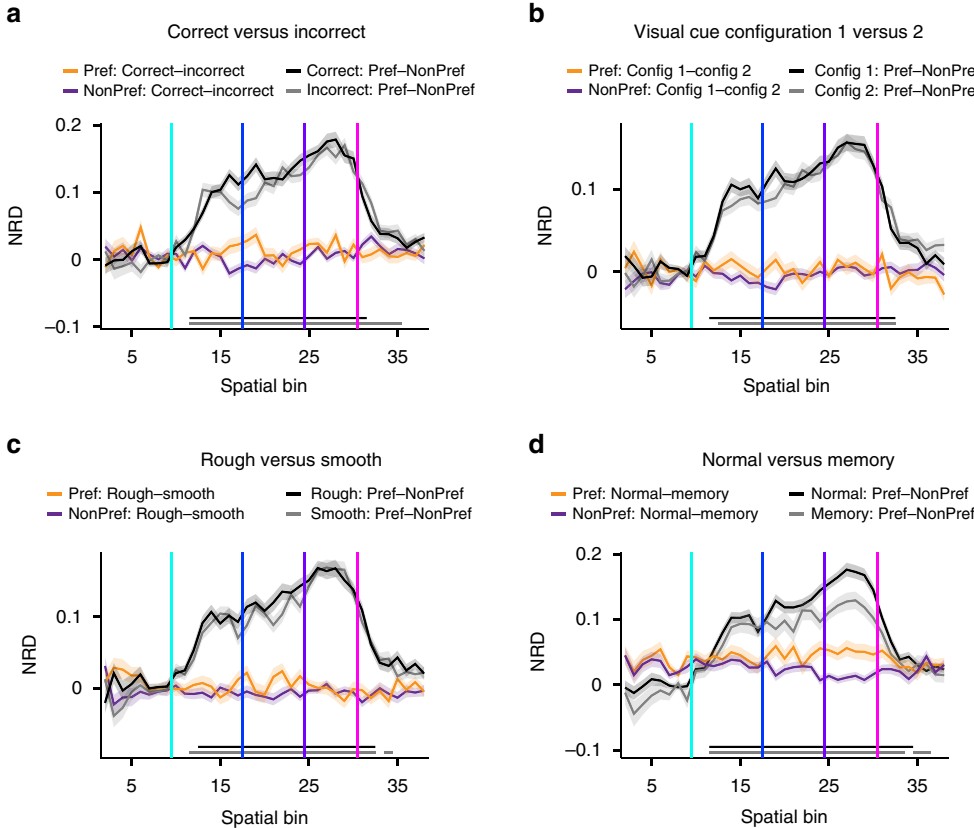

**Figure 4 | Control analyses to determine whether the sustained pattern of discriminatory firing in PRH can be ascribed to various task variables.**
(**a**) NRDs between correct and incorrect trials, averaged across the total PRH population and plotted as a function of spatial location. Vertical coloured lines correspond to the spatial decision point (cyan), PNR (blue), reward site (purple) and return to middle (pink). The difference between correct and incorrect trials is rendered in orange when the comparison was made for the preferred (Pref) arm of each cell, and in purple for the non-preferred (NonPref) arm. When all correct trials were split between those made into the preferred versus non-preferred arm (black curve), a high left–right selectivity was found throughout the arm visit. The same held for the incorrect trials (grey curve). (**b**) Same as **a**, but now for the identity of the visual cues shown on the screen that were positioned to the left versus right of the animal when its head was oriented forward in the middle lane. Again, no significant difference is observed when visual cue configurations are contrasted within the subset of preferred-arm visits (orange) or the subset of non-preferred arm visits (purple). In contrast, a clear difference between preferred and non-preferred arms is seen both in the subset of trials with configuration 1 (Config 1; black) or that with configuration 2 (Config 2; grey). (**c**) Same as **a**, but now for rough versus smooth sandpaper textures, which covered the side walls from the decision point to the PNR. (**d**) Same as **a**, but now for normal trials versus memory trials. Horizontal lines below the four curves of each graph indicate statistical significance with respect to all baseline bins (baseline = bins 3–6; Wilcoxon signed-rank test, $\alpha = 0.01$); the colour code of these lines is the same as for the firing-rate difference curves (no significant effects were found for comparisons within the same arm type).

component, as defined relative to baseline, both of which contribute to strong discrimination.

We next asked whether the sustained discriminatory firing of PRH neurons can be ascribed to the global positioning of the animal (that is, in the left or right arm), to task variables that are associated with the animal's choice behaviour, or with requirements to complete a correct trial. For instance, animals may maintain stimulus information when passing the PNR, until they reach the reward site and return to the middle lane. We first tested whether the sustained PRH response may cohere with completing a correct, rewarded trial. Reward delivery requires not only a correct initial choice but also the prolongation of the locomotor response beyond the PNR, which may be coupled with a higher reward expectation than during an incorrect trial. This test failed to produce a significant correlation with sustained PRH responses (Fig. 4a). When we contrasted correct versus incorrect trials for the preferred arm alone, we found no discriminatory firing. The same applied for the non-preferred arm. However, when contrasting the preferred versus non-preferred arm for correct trials only, strong discriminatory firing was found, and the same held for incorrect

trials. Similarly, the presentation of the CS + stimulus on the left versus right of the middle lane did not correlate with the generation of sustained PRH responses, as was also the case for the CS − stimulus (Fig. 4b). Likewise, the identity of somatosensory cues along the initial segments of the arms did not explain the selectivity of PRH firing patterns (Fig. 4c). Whereas in the main paradigm described so far the CS + and CS − images were displayed until the rat passed the PNR, we interleaved memory trials having the image offset occur 2 s before the front barrier was removed in advance of the animal's initial choice. Although the memory load in these trials was higher than in normal trials, they did not reveal a change in left–right discrimination (Fig. 4d).

In addition to global positioning in the task environment, sustained PRH responses may arise from left versus right trials being accompanied by different spatial views of the environment (scene perception) or by different directions of rotational movement (coupled to sustained vestibular input). Furthermore, a left run can be argued to represent a different phase of the behavioural task than a right run, and not merely a different large-scale positioning. To assess which of these possibilities

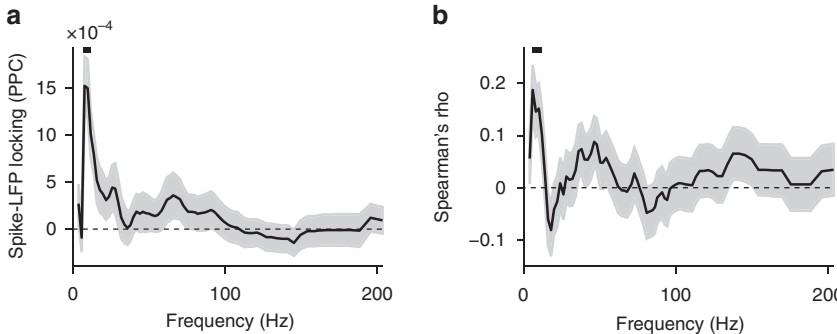

**Figure 5 | Selectivity of PRH neurons correlates with phase-locking to hippocampal theta oscillations.** (**a**) Mean ( ± s.e.m.) of spike-LFP locking (PPC) of PRH cells to CA1 theta oscillations as a function of LFP frequency. PPC is pairwise phase consistency. (**b**) Pairwise phase consistency and discrimination score of all PRH neurons were selectively correlated in the hippocampal theta band. Spearman's rho correlation between s.e.m.'s was computed using jack-knife estimation. Vertical bar at top indicates significance at $\alpha = 0.05$.

applied to PRH responses, we contrasted trials in which the animal first passed the PNR, collected a reward and subsequently moved forward or travelled in the opposite direction on the same individual arm, up to the PNR but no further. Here runs in the opposite direction represent a deviation from the regular task sequence. PRH cells that were left- or right-arm selective during regular forward navigation expressed a similar selectivity when the animal travelled in the opposite direction (Fig. 3l; $N = 349$, Pearson's correlation coefficient $r = 0.62$: $P = 1.56 \times 10^{-38}$). Thus, sustained PRH responses are best explained by a large-scale, segmental selectivity, not by other task or sensory variables. Finally, the residual discriminatory firing occurring after animals returned to the middle lane (Fig. 2e) was also observed in the spatial domain (Figs 3a,d and 4, and Supplementary Fig. 2i,k). This retention effect could not be explained by behavioural trajectory differences of rats having passed the PRM (Supplementary Fig. 2j,l), suggesting a form of retrospective coding.

Considering that spatially extended firing fields of PRH neurons seem to co-localize with branch points, whereas dorsal CA1 place fields may be more scattered across the maze (Fig. 3), we quantitatively tested this contrast by determining the start and end points of firing fields of cells in both structures (the start point is defined here as the point of entry into the firing field according to the animal's heading direction during task performance; Supplementary Fig. 5). First, we observed that the circular variance of the PRH start and end points was significantly lower than for CA1, indicating that indeed the spatial scattering of PRH firing field boundaries is less than for CA1 (Supplementary Fig. 5e and Supplementary Table 3; all relevant CA1–PRH comparisons were significant using a bootstrap test). Next, we compared, on one hand, the distances between the start points of the firing fields and the decision point (cf. Fig. 1a,b) and, on the other hand, the distances between the end points and return branch point (Supplementary Fig. 5f,g; see Methods). For most comparisons between PRH and CA1, the firing fields of PRH cells were indeed better aligned with these two branch points than place fields of CA1 cells were. Especially the end points of firing fields from both the preferred and non-preferred arms of PRH cells were more closely aligned to the return branch point than those of CA1 cells (bootstrap test). Furthermore, the start points of PRH firing fields on the non-preferred arm were closer to the decision point than those of CA1 cells (bootstrap test). The only PRH–CA1 comparison failing to reach significance was between the start-to-decision point distances of PRH cells on the preferred arm versus CA1 cells (Supplementary Fig. 5f,g and Supplementary Table 3). Despite the impression raised by Fig. 3d (which shows NRD, a contrast measure),

this lack of effect was due to relatively many PRH activation patterns surpassing threshold in between the decision point and reward site (Supplementary Fig. 5b; see also Fig. 3e). In agreement with the strong contribution PRH deactivations make to side specificity, the PRH fields coded by decrements in firing rate were more tightly locked to branch points than the increments.

**Interactions between PRH cortex and hippocampus.** Within the sensory-to-hippocampal hierarchy[16,24,35,36], selective firing of PRH cells may arise in a bottom-up fashion where S1BF and other neocortical areas feed sensory inputs into PRH, which subsequently enables the hippocampus to sculpt a refined, spatial coding. We observed, however, that firing selectivity began to increase around the same point in the task (that is, the spatial decision point, leading up to the PNR) in PRH, CA1 and sensory cortices (Figs 2e, 3 and 4, and Supplementary Fig. 3) and similarly decreased around PRM. This prompted us to examine whether rhythmic dynamics of processing in either sensory neocortex or hippocampus may play a coordinating role in the selectivity of PRH responses. Specifically, we studied the phase-locking of PRH firing patterns to rhythmicity in local field potentials (LFPs), which provides a measure of phase synchronization between local firing activity and synaptic mass activity reflected in the LFP, indicative of direct or indirect communication. First, we found a significant phase-locking of PRH cells selectively to the theta band of hippocampal LFP (Fig. 5a). Second, the discrimination score of PRH cells was significantly positively correlated with the strength of phase-locking to hippocampal theta-band activity (Fig. 5b). We further examined whether this positive correlation between PRH cell selectivity and hippocampal theta was found for S1BF or CA1 units, but this was not the case. S1BF showed a weak positive correlation with hippocampal gamma-band activity, whereas CA1 units showed a weak negative correlation to hippocampal theta- and gamma-band activity. We also analysed the relationship between the discrimination score of PRH cells and LFP activity recorded from S1BF, but note that this LFP activity may well be contaminated by volume-conducted hippocampal LFP activity[37]. In conclusion, the selectivity of PRH neurons is specifically associated with the type of hippocampal dynamics that predominates during spatial navigation and expression of place-cell activity[4,5].

## Discussion
These results may have significant implications for our understanding of the genesis of contextual and spatial coding in the hierarchy from sensory cortices to the hippocampal system.

First, our findings support a wider view of PRH than being merely implied in the processing of single objects. The firing patterns of PRH neurons, integrating across large maze segments, are consistent with the previously proposed role for PRH in discrimination of complex configurations of visual stimuli[30], contextual and spatial discrimination[26,38], and spatial scene learning[39]. As indicated by our results, PRH coding of such configurations may assume the form of larger spatial-contextual aggregates or environmental 'chunks', such as one of the side arms or the middle lane. This PRH coding is distinct from that of hippocampal place cells, underscoring that hippocampus and PRH are not part of one mass-action unit[30] (cf. ref. 40). In conjunction with the fine-grained spatial coding provided by dorsal entorhinal grid cells or hippocampal place cells, PRH neurons may be especially useful in coding a pointer to the animal's large-scale position in its task environment (as delineated by spatial decision points), which may subserve ongoing spatial navigation, path planning, decision-making and memory formation.

In terms of firing dynamics, it was previously shown that whole-cell-recorded PRH neurons in vitro generate prolonged spike trains induced by current pulses or synaptic inputs[41] and, in agreement with these findings, we show how these intrinsic properties are compatible with a function in the sustained neural coding of 'chunks' of maze trajectories. Along with sustained activation of PRH cells in one arm of the maze, we typically found sustained loss of firing of the same cells in the other arm. This deactivation may correlate either with a loss of excitation or with a powerful feedforward inhibition, which has been shown to control PRH firing in isolated brains and may serve to gate the bidirectional transfer of information between hippocampus and neocortex[42]. These or related inhibitory mechanisms may effectively enhance coding of maze chunks by suppressing activity on one side of the maze but permitting signal transmission on the other. However, the precise mechanisms underlying discrimination between maze segments awaits further investigation. Here the main point worth emphasizing is that PRH cells are capable not only of making a binary spatial distinction but also of making graded distinctions between multiple maze segments (for example, left versus right versus middle).

It remains to be determined whether the environmental chunks coded by PRH neurons are truly spatial (in the sense of representing positional information allocentrically), and indeed the role of proximal versus distal cues and contextual elements in shaping this code awaits further investigation. Nonetheless, the current results show that task-related variables, such as sensory cue configurations, and correctness of behavioural performance, do not explain left–right discriminations. Because reward was administered only in correctly performed trials, discriminatory firing is unlikely to be explained by differences in its delivery or expectancy. In addition to the strong discrimination PRH neurons made between the left and right arm, a more subtle residual left–right discrimination was observed after rats returned to the middle lane. The exact cause of this effect is unknown, but is unlikely related to rewards applied in the middle lane because it remained visible when the left versus right arm trials were contrasted given correct or incorrect performance. In other words, even when no middle-lane reward was provided, a left–right difference was observed. Our data further suggest that residual discriminatory firing, occurring when rats had returned to the middle lane, is an effect of trial history and reflects retrospective coding (that is, a dependence of PRH firing on where the rat came from in the foregoing trial). Although it is notable that this retrospective activity occurred under different behavioural conditions, it remains to be examined what the main factors are for determining the strength of this effect.

Furthermore, the strong correlation in left–right selectivity between forward and opposite-direction runs occurring after the rat had passed the PNR speaks against a predominant influence of head and eye orientation, of single objects in the environment captured within a spatial view, or vestibular cueing. Indeed, the finding that left–right selectivity was sustained across entire arcs, encompassing multiple spatial views on the environment, strongly argues against a purely object-oriented function of PRH coding, while being compatible with a unitizing function. That PRH neurons are distinct in this type of chunking is underscored by the more punctate and transient firing patterns found in hippocampus and sensory neocortex. Thus, although more work is needed to flesh out the exact determinants of PRH representations, the current findings outline a unique role of PRH in contextual representations.

Previous unit-recording studies in freely moving rats emphasized a role of PRH in coding object-related information[18,34,43], raising the question why these did not identify sustained firing that was selective for large maze chunks. Two key differences with the present study are that these previous experiments studied rat navigation in a single spatial compartment, and that they did not use tasks in which the animal needed to apply knowledge of its large-scale position to expect and secure reward. The plausibility of sustained PRH responses being constrained by behaviourally significant locations (such as the spatial decision point) is underscored, first, by the strong input this area receives from prefrontal and anterior cingulate areas[44–46], which have been implicated in coding task rules, choice parameters and attentional set[47–50]. Second, the selectivity of PRH neurons was most prominently expressed between two maze locations separating distinct task choices and paths: the decision point and return to middle.

Although PRH firing behaviour could only be compared directly to three brain areas including dorsal area CA1, it is relevant to relate it to other regions. Ventral CA1 neurons express large place fields, which appear as scaled-up versions of fields found in dorsal CA1, and thus likely differ from PRH neurons in that they are scattered across the rat's environment, not showing a demarcation by spatial decision points or other behaviourally significant landmarks[9,51]. This larger spatial scattering was confirmed for our dorsal CA1 recordings compared to PRH neurons. Furthermore, the start and end positions of PRH firing fields were generally better aligned to the branch points on the maze (the start points of PRH fields on the preferred arm presented an exception). The left–right selectivity of PRH neurons is also somewhat reminiscent of rat parietal neurons. A number of studies suggest that during freely moving behaviour, rat posterior parietal cortex predominantly shows—often transient—firing patterns correlated to the animal's progression through a route, including performance of particular behaviours (for example, body turns) and movement direction (for example, refs 52,53), whereas PRH coding is marked by more sustained firing coupled with a large spatial extent of firing fields (current study). Nonetheless, also parietal neurons can code information on the rat's spatial position and may change firing rate during prolonged stretches on a maze[52–54]. Thus, these suggested differences await further testing in a direct comparison by simultaneous posterior parietal cortex and PRH recordings during maze behaviour. Regarding the subtle but significant retention effect in PRH observed after the rat's return to the middle lane, this type of retrospective coding is similar to that found in CA1 and entorhinal cortex neurons, at least under some task conditions such as spatial alternation[55]. However, we did not observe retrospective coding in area CA1 of rats performing the visual discrimination task. Also prospective coding of future paths has been reported in area CA1 of rats performing spatial

alternation[55,56], but this was not found in PRH or CA1 during our visual discrimination task.

The sustained nature of firing brings up the question whether PRH neurons may encode working memory operations, as has been reported for instance in rat medial prefrontal cortex[57]. Several results argue against the (classic) idea of working memory being a main function of the PRH coding we report here. First, whether or not a trial included a memory component did not matter for PRH discriminatory firing. Second, a correlate of working memory would be expected to be in place until the relevant action and reward acquisition have occurred, but not afterwards. In contrast, we found that the sustained firing patterns continued well after the reward sites had been visited, declining when the rat returned to the middle lane. Third, after passing the PNR, no working memory was required to successfully complete the rest of the trial, yet the PNR-to-PRM trajectory was marked by a pronounced sustained firing. Finally, even when the rat deviated from the regular trial sequence and walked in the opposite direction on an arm, spatially selective firing remained intact. However, in a more general sense the PRH code may serve to keep track of the animal being localized on a large maze segment during task performance.

A subject of future investigation is how PRH computes its spatially selective responses, and which afferent sources provide the information to do so. While PRH sends a direct projection to the hippocampus[44,58] and PRH cells could thus affect place fields in CA1, our findings highlight the entrainment of PRH cells to the hippocampal theta rhythm. Given the evidence for sparse and tightly controlled communication between PRH and the hippocampal–entorhinal complex[42,46], the correlation between the contextual selectivity of PRH neurons and their phase-locking to hippocampal theta suggests that hippocampal activity may be critical for opening up communication channels for transmission of PRH signals to a range of target structures. In this context, the hippocampus may exert a top-down influence on PRH via outputs from CA1 and subiculum[44,59]. In addition, afferent inputs from postrhinal cortex could contribute to sustained PRH patterns, as this area has been implicated in visuospatial processing and scene perception[44,60].

Altogether, the current findings reveal a role of PRH neurons in coding large chunks of an animal's task environment, contrasting with more transient, local response patterns of hippocampal and sensory cortical neurons. Because individual task elements, such as auditory cues, are co-represented in PRH with large spatial segments, even by the same cells, the findings argue against a rigid separation of what versus where processing in the medial temporal lobe (cf. ref. 15) and favour an integrative function of PRH, combining chunking of maze segments with coding of individual items. From a conceptual viewpoint, both maze chunks and other items could be argued to be objects, defined as composites of informational elements bound by common multisensory and spatiotemporal properties[61]. Thus, the current findings do not contradict a role of PRH in coding what information *per se*. Hippocampal theta rhythm may provide a means for temporally coordinating large-scale and more refined coding by PRH and dorsal CA1 cell populations, respectively. The coding characteristics of PRH cells reported here are consistent with lesion studies in primates and humans[6,7], pointing to an important role of the parahippocampal gyrus (including PRH) in topographical orientation abilities.

## Methods

**Subjects.** Data were collected from three male Lister Hooded rats, 28–46 weeks of age (cf. ref. 37). During handling and behavioural training, animals were communally housed in standard cages under a reversed day/night cycle (lights off: 8:00; lights on: 20:00). During behavioural training and the main experiment,

animals were food-restricted to maintain their body weight at 85% of free-fed animals, taking the *ad libitum* growth curves of Harlan www.harlan.com/online_literature/research_models.hl) and the growth curves in ref. 62 as references. From 2 days before surgery until after a full post-surgery recovery week, food was available *ad libitum*. Rats had *ad libitum* access to water during all phases of the experiment. After surgery, animals were housed individually in transparent cages ($40 \times 40 \times 40$ cm). All experiments were conducted according to the National Guidelines on Animal Experiments and were approved by the Animal Experimentation Committee of the University of Amsterdam. Animals were not subjected to differential treatment and thus no blinding procedure was applied.

**Apparatus and stimuli.** The animals were trained on a two-choice visual discrimination task set on a figure-8 maze (Fig. 1a; $114 \text{ cm} \times 110 \text{ cm}$). The paths of the maze were 7 cm wide (that is, 7 cm separation between walls) and were flanked by walls that were 4 cm high. The floor of the maze was elevated 40 cm above ground level. During the inter-trial interval, the rat was confined to the middle lane of the figure-8 maze using two movable Plexiglas blocking walls, referred to as the front and back blocking walls (with the front wall closer to the display screens). Stimuli used were two equiluminant figures (inverse Wingdings; Microsoft, Redmond, WA; either a diamond or plane; Fig. 1a) that had the same proportions of black and white pixels. The stimuli were simultaneously presented on two monitors (LCD, Dell, 15 inch). At the sides of the arms of the maze into which the rat entered after the decision point, strips of rough (P40) or smooth sandpaper (P180) were glued to the inner walls of the maze (red strips in Fig. 1a). Food pellets were delivered at three reward sites, one of which was placed in the middle lane (ITI confinement space) close to the front blocking wall. Rewards were delivered only after the animal had passed the PNR. The other two reward sites were situated in the left and right arms, respectively (Fig. 1a). Eight photobeams were attached to the outer walls of the maze, with two photobeams in the middle lane and three photobeams per side arm. Two of these photobeams were positioned to record the rat passing the PNR and PRM. The behavioural programme was controlled using Matlab. Events that were detected by the behavioural apparatus and the commands issued by the behavioural programme were directly time-stamped and synchronized with electrophysiological data by feeding them as inputs into the Neuralynx data acquisition system.

**Behavioural procedure.** Following an inter-trial interval with a duration randomly selected between 15 and 25 s, the two-choice visual discrimination task began at the onset of a 2 kHz sinusoid sound cue that lasted for 0.1 s. At this sound, the rat could break an infrared photobeam in the middle-lane confinement space close to the front blocking wall, causing two visual stimuli to appear. For each rat, one of the stimuli was designated the CS + stimulus, while another stimulus was designated the CS − stimulus. In every trial, both the CS + and CS − were presented, with the spatial location (screen to the right or left of the middle lane) of the stimuli varying randomly. The CS + and CS − stimuli were shown on the left and right for equal amounts of trials. Using pseudo-random sequences we made sure that for each series of 10 consecutive trials, 5 had the CS + presented on the left and 5 had the CS + on the right side. Throughout the session no more than three same-side trials were applied in succession.

The front blocking wall was removed 4 s after stimulus onset. The rat could now enter one of the two side arms at a location labelled the 'decision point' (Fig. 1b). In a correct trial, the rat chose the side at which the CS + was presented; choosing the other side failed to produce a reward. The rat's final choice was indicated by the animal breaking the infrared photobeams that were positioned beyond the visual screens at the end of the sandpaper walls (Fig. 1a,b; PNR). On passing this point, the stimuli were immediately turned off and the rat could no longer correct its initial choice by walking back and entering the other arm. At any point in a side arm beyond the PNR, however, the rat was able to turn around and walk in the direction opposite to the regular forward path until reencountering the (now blocked) PNR again. We refer to these trials with late stimulus offset as normal trials. We also included a set of memory or early-offset trials, which comprised, on average, 30% of all trials in a session. During these trials, stimuli were displayed for only 2 s, implying that the rats had no access to the visual stimulus for 2 s before they were allowed to make a choice. On a correct choice in either type of trial, two or three pellets (BioServ, dustless precision pellets, 14 mg) were delivered in the ceramic cup that was located in the same arm. The conjunction of a correct choice and rough sandpaper resulted in three pellets, while a correct choice and smooth sandpaper resulted in two pellets. When the rat returned to the middle lane after making a correct choice (but not after making an incorrect choice), an additional pellet was placed in the middle lane cup, and the front and back blocking walls were placed back in position, after which the next inter-trial interval commenced. The median time between return to the middle lane and the initiation of a new trial was 5.3 s. During recording sessions, rats performed $60.1 \pm 2.1$ (mean ± s.e.m.; $N = 46$ sessions) trials. Performance was significantly better than chance level ($59.2 \pm 0.5\%$ correct, mean ± s.e.m.; $P = 2 \times 10^{-8}$, Wilcoxon's signed-rank test). While this performance level may seem modest, it should be emphasized that several features made the task relatively difficult for rats: (i) the isoluminance of the stimuli, preventing a choice strategy based on contrast; (ii) the relatively distal placement of the visual presentation screens, about 45 cm away from the rat, with

an inter-screen distance of 25 cm. Moreover, well-known behaviours such as trial-to-trial alternation may have affected performance.

**Surgical procedure and recording drive.** Each rat's right hemisphere was implanted with a custom-built microdrive (labelled here as quad-drives; cf. refs 37,63) containing 36 individually movable tetrodes, including eight recording tetrodes directed to the PRH cortex (area 35/36; skull coordinates: $-5.0$ mm anteroposterior (AP) and 5.0 mm mediolateral (ML)[64], eight to the dorsal hippocampal CA1 area ($-3.5$ mm AP and 2.4 mm ML), eight to the somatosensory cortex (S1BF: $-3.1$ mm AP and 5.1 mm ML) and eight to the visual cortex (V1M, $-6.0$ mm AP and 3.2 mm ML to bregma, with one additional tetrode per area that could be used as a reference). The quad-drive weighed 23 g and was about 52 mm in height. The PRH bundle was placed at an angle of 17° pointing laterally and an angle of 24° pointing caudally with respect to a perpendicular orientation relative to the skull, such that tetrodes were aimed at area 35 and 36 (as a reference we used the border of 35–36 area indicated in ref. 64: $-6.0$ mm AP, 6.4 mm ML and $-6.2$ mm ventral to bregma). Before surgery, rats received a subcutaneous injection of buprenorphin (Buprecare, 0.01–0.05 mg kg$^{-1}$), meloxicam (Metacam, 2 mg kg$^{-1}$) and Baytril (5 mg kg$^{-1}$). Rats were anaesthetized using 3.0% (induction) and 1.0–3.0% (maintenance) isoflurane. Animals were mounted in a stereotaxic frame and body temperature was maintained between 35 and 36 °C using a heating pad. After the cranium was exposed, six holes were made to accommodate surgical screws. Four holes (each ∼1.8 mm in diameter) were drilled for placement of the four bundles holding the tetrodes. After removing the dura, the bundles were lowered onto the exposed cortex and the quad-drive was fixed to the skull and to the surgical screws using dental cement. One of the skull screws located in the caudal part of the parietal skull bone contralateral to the drive location served as ground. After having thus anchored the drive, the tetrodes were lowered 0.4–1.0 mm (depending on the target area) into the cortex. Over the next 7 days, the animal was allowed to recover, with ad libitum food and water available. The recording and reference tetrodes were gradually lowered to their target region over the course of the first 8–10 days after implantation, with their depths recorded daily. Depths were estimated by the number of turns of the guide screws and by online monitoring of LFP and spike signals.

**Histology.** After the final recording session, current (12 µA for 10 s) was passed through one lead per tetrode to mark the end point of the tetrode with a small lesion. Twenty-four hours later animals were deeply anaesthetized with Nembutal (sodium pentobarbital, 60 mg ml$^{-1}$, 1.0 ml intraperitoneal; Ceva Sante Animale, Maassluis, the Netherlands) and transcardially perfused with a 0.9% NaCl solution, followed by a 4% paraformaldehyde solution (pH 7.4, phosphate-buffered). Following post-fixation, transversal sections of 40 µm were cut using a vibratome and stained with Cresyl Violet to reconstruct tetrode tracks and localize their end points. Recording locations were carefully reconstructed using the end points and the recorded number of turns.

**Data acquisition and spike sorting.** Using tetrodes[65] (Nichrome wire, California Fine Wire, diameter: 13 µm, gold-plated to an impedance 500–800 kΩ at 1 kHz), we recorded neural activity with a 144-channel Digital Neuralynx Cheetah set-up (including 16 reference channels; Neuralynx, Bozeman MT). Signals were passed through a unity-gain pre-amplifier headstage, a 144-channel, automated commutator (Neuralynx) and bandpass filtered between 600 and 6,000 Hz for spike recordings. One millisecond epochs of activity from all four leads were digitized at 32 kHz if a signal on any of the leads of a tetrode crossed a pre-set voltage threshold. LFPs recorded on all tetrodes were continuously sampled at 2,035 Hz and bandpass-filtered between 1 and 500 Hz. Spike trains were sorted to isolate single units using a semi-automated clustering algorithm followed by manual refinement (KlustaKwik, by K. Harris, and MClust 3.5, by A.D. Redish). Automated and manual clustering of spikes was performed using the waveform peak amplitude, energy and first derivative of the energy. Clusters were accepted as single units when having no more than 0.1% of inter-spike intervals shorter than 2 ms. This criterion was equally applied across all four recorded regions. During recordings, rat behaviour was video-tracked at 25 Hz, and an array of light-emitting diodes on the headstage allowed offline tracking of the rat's position. Estimates of sample size (cell counts) were based on the prior literature, for example, on hippocampal place cells.

**Statistical testing of neural correlates.** For statistical testing of individual response curves in the peri-event time histograms (for example, Fig. 2 and Supplementary Fig. 2), firing rates of left- and right-sided trials were tested against a baseline taken from the ITI preceding trial onset (Wilcoxon signed-rank test). Although this interval is commonly used in the electrophysiological literature to compute a baseline, it should be emphasized that in view of the type of spatial selectivity of PRH cells, this choice of baseline is essentially arbitrary (cf. Fig. 2b). To compensate for multiple comparisons, each bin of interest was tested against eight, contiguous baseline bins in a late time segment of the inter-trial interval, at $-3$ to $-1$ s relative to the auditory sound at trial onset (bin width: 250 ms). To qualify as significant, the P values of a test bin needed to exceed the significance

threshold ($\alpha = 0.05$) against each baseline bin. Thus, the $\alpha = 0.05$ criterion had to be met eight times in succession (cf. refs 63,66).

To assess the statistical significance of differences between two conditions (for example, left versus right) we also made comparisons per bin, using Mann–Whitney U-test. P values were false discovery rate-corrected for multiple comparisons ($\alpha = 0.05$). Both correct (rewarded) and incorrect (non-rewarded) behavioural trials were included in the analyses described here and below, unless noted otherwise. Trials in which the rat would turn around before reaching the PNR and choose the other arm were rare (on average <1 per session) and were also included.

**Spatial linearization of the maze.** The figure-8 maze was segmented into a total of 51 spatial bins, made up of 21 spatial bins for each side arm and 9 bins for the middle lane. The decision point corresponded to bin 9 and the return branch point to bin 1. In the spatial domain we used the return branch point as reference because of its topological significance, whereas the PRM photobeam (located at bin 3) was only used to document the animal's behaviour in the time domain (Figs 1 and 2). Bins were ∼9 cm in length. The rat's position, read out from a given video frame, was assigned to a spatial bin using the minimal Euclidean distance between the rat's position to the bin centre positions. Video frames from trials in which the animal took more than 15 s to respond following stimulus onset (that is, until breaking a PNR beam) were excluded.

**Statistical measure of spatial discrimination.** To derive an unbiased metric of the degree to which firing rates of recorded neurons discriminate between visits to the left and right arms of the maze, we computed a discrimination score as follows. We first computed the actual absolute difference in firing rate between left and right trials for a given time point, called $D_a$. The range of time points included in this computation were dependent on the chosen synchronizing event (cf. Supplementary Fig. 3). Next, we performed 100 permutations of left and right trials and computed the difference in absolute firing rate between the two conditions after shuffling, called $D_s$. We then computed the s.d. of the distribution of $D_s$ values, and computed a T-statistic $T_a = (D_a - D_s)/\text{s.d.}$, which we refer to as discrimination score. This Z-statistic attains statistical significance with values $>1.63$. Two criteria were set for the inclusion of trials: (i) the rat was required to emit a response by passing the PNR within 15 s after stimulus onset; (ii) each condition needed to contain at least five trials. The durations of time periods during which the discrimination score was significantly elevated (that is, larger than 1.63 during side-arm visits; cf. Supplementary Fig. 3) were compared between brain areas using Kruskal–Wallis and post hoc Tukey–Kramer tests with a 95% confidence interval.

**Preferred arm and the NRD.** Some PRH units showed a firing-rate increase in one particular arm of the maze, while others showed a decrease for that same arm. If one would take an average firing rate for one arm across all units, the different types of firing-rate modulation would cancel out. For this reason we calculated, for each unit, which of the two arms was 'preferred' in terms of being correlated to its highest firing rate. First, we normalized the firing rates of each unit, assigned across all 51 spatial bins, by dividing these values by its maximal firing rate. The maximal rate of each unit was selected from all bins from both the left and right arm trials. Next, we calculated the average normalized firing rate for all middle-lane bins, as well as all 21 left- and all 21 right-arm bins, using the time periods associated with forward locomotion. Normalized firing rates were only determined for units with an average discrimination score $>0.5$ calculated across the entire side arms (this criterion results in different single-unit counts than listed in Supplementary Table 1). If the mean normalized firing rate for a given unit was higher for the left arm than for the right arm, the left arm was designated as 'preferred arm' and the right arm as 'non-preferred arm' and vice versa. For each unit, we next computed the NRD in the spatial domain by subtracting the normalized firing rates of the spatial bins of the non-preferred arm from the normalized firing rates of the corresponding bins of the preferred arm. The same procedure was followed when computing NRD values in the temporal domain, now taking temporal instead of spatial bins. Left–right firing-rate differences for locomotion in forward and opposite directions (Fig. 3l) were determined between the reward site and PRM.

In the time domain, the significance of mean NRDs was tested against 2 s of contiguous baseline bins (10 ms in width; see also above, Statistical testing of neural correlates). If a bin was significantly different from all baseline bins it was taken as significant ($\alpha = 0.01$).

To calculate whether single units significantly differentiated between the left and the right arms, we compared the time segments of left- versus right-sided trials corresponding with the rat's occupancy of the side arms ($-1$ s to $+5$ s around PNR and $-5$ s to 0 s around PRM, Mann–Whitney U-test, $\alpha = 0.05$).

**Spatial consistency of left and right differences.** Whereas PRH neurons generated sustained and spatially extended firing patterns distinguishing between the left and right arms, hippocampal and sensory cortical neurons generated more phasic differential patterns (for example, Fig. 2). To quantify this difference, we used a measure of the consistency of firing-rate differences between the left or right arm across spatial bins. For each cell, we took the NRD across the spatial bins

corresponding to the side arms (bins 12–30), which in almost all cases yielded positive NRD values. NRD values smaller than zero were set to 0. We then ensured that again for each cell, the NRD values summed up to 1, by dividing NRD(x) by the sum of NRD(x) over the 19 bins (where x represents spatial bin number), thus defining a quasi-probability variable p. We then computed the entropy of NRD by summing up −p(x).log(p(x)) over all bins, and dividing this sum by log(19), corresponding to the 19 spatial bins. This yields a measure of spatial consistency ranging between 0 and 1. If the NRD is constant across spatial bins, corresponding to a situation where left–right differences are constant across all spatial bins, then the consistency is maximal (1), whereas if the NRD is non-zero only at a single bin, the consistency is minimal (0).

**Distribution of start and end points of firing fields.** To quantify the distribution of spatial start and end points of sustained PRH firing fields and compare these to CA1 place fields, we divided preferred and non-preferred trial trajectories into 30 bins, made up of 9 middle lane bins and 21 side arm bins (Supplementary Fig. 5a). As above, this analysis included cells with an average discrimination score > 0.5, and only forward locomotion periods were taken into account. For each PRH unit the baseline was calculated by taking the average firing rate across all preferred and non-preferred bins. For both the preferred and non-preferred side this baseline was subsequently subtracted from the binned firing rates. Because PRH neurons show deactivations on their non-preferred side, we multiplied these values by − 1 to obtained absolute values of deflections from baseline. Sustained activity was subsequently determined by a method similar to those used to identify CA1 place fields[67,68]. First, the data were smoothed using a five-point Gaussian. Second, we determined the longest stretch of contiguous bins with a minimal firing rate of > 20% of the maximum rate. A PRH neuron was designated as having an extended firing field if the number of contiguous bins reaching > 20% was at least nine. Its start point was defined as the first bin (according to the animal's locomotion direction during task performance) surpassing the threshold, while the end point marked the last bin where this threshold was reached. CA1 neurons generally have smaller place fields. For these cells, at least three adjacent bins were required to include them in the analysis. For the purpose of determining the start and end points of spatially extended firing fields, the route travelled in a single, full trial was assumed to be circular, so bin 1 was adjacent to bin 30. The assumption of circularity has the advantage that start and end point estimates are not biased by arbitrary cut-offs.

We determined the amount of scattering of firing fields across the maze by computing the circular mean and circular variance of their start and end locations (circular statistics toolbox, MATLAB[69]). 95% Confidence intervals of the variance in start and end points were calculated by bootstrapping these points 1,000 times within each condition and finding the outer 2.5 percentiles. Non-overlapping confidence intervals are thus significantly different at α = 0.05 (Supplementary Fig. 5). Subsequently, the alignment of the start and end points of firing fields with branch points was determined by calculating the spatial offset (in bins) between a cell's start point and the decision point (bin 9; Fig. 1a,b) and the offset between a cell's end point and the return branch point (bin 1).

**Spike to LFP phase-locking.** For every frequency f, we determined the spike-LFP phases by cutting out LFP segments of length 7/f s (that is, 7 cycles) centred around each spike. Spikes were exclusively related to LFPs recorded from a different electrode to avoid contamination of the LFP by the spike itself. The spike-LFP phases were then obtained as the complex arguments of the Hann-tapered LFP segments.

The strength of spike-LFP phase-locking (Fig. 5) was quantified by the pairwise phase consistency (PPC), which is unbiased by the number of spikes[70]. For the j-th spike in the m-th trial we denote the average spike-LFP phase as $\theta_{m,j}$, and similarly for the k-th spike in a different trial l, where dependence on frequency is omitted in what follows. The PPC is defined as

$$\hat{\psi} = \frac{\sum_{m=1}^{M} \sum_{l \neq m}^{M} \sum_{j=1}^{N_m} \sum_{k=1}^{N_l} \cos(\theta_{l,k} - \theta_{m,j})}{\sum_{m=1}^{M} \sum_{l \neq m}^{M} N_m N_l},$$

where $N_m$ is the number of spikes for the m-th trial. The PPC quantifies the average similarity (that is, in-phaseness) of any pair of two spikes from the same cell in the LFP phase domain. Note that all pairs of spikes from the same trial are removed by virtue of preventing $l = m$ in the above equation, because spike phases from the same trial can typically not be treated as statistically independent random variables[70]. PPC values were averaged across the different hippocampal electrodes, and were computed in the 5–10 s window after image onset, and then correlated on a cell-by-cell basis with the discrimination scores computed in the same window. Spearman's correlation was used to avoid assumptions of parametric regression testing.

**Data availability.** All relevant data are available from the authors.

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

## Acknowledgements

We are grateful to Conrado Bosman, Carien Lansink, Umberto Olcese and Jeannette Lorteije for their comments on the manuscript, and acknowledge the software tools provided by Kenneth Harris (University College London; KlustaKwik) and by A. David Redish (University of Minnesota, Minneapolis; MClust). We are grateful for the support provided by the Technology Center of the University of Amsterdam, in particular by Ron Manuputy, Harry Beukers, Gerrit Hardeman, Ed de Water, Eric Hennes and Matthijs Bakker. This work was supported by the Netherlands Organization for Scientific Research-VICI Grant 918.46.609 and Netherlands Organization for Scientific Research ALW grant 820.02.020 (both to C.M.A.P.).

## Author contributions

J.J.B., M.V. and C.M.A.P. designed and planned the experiments and analyses; J.J.B. and M.V. performed the quad-drive tetrode recordings, with support from L.A.v.M.-D. and J.C.J.; J.J.B., M.V., C.M.A.P. and M.P.W. analysed the data; C.M.A.P., J.J.B. and M.V. wrote the paper in collaboration with the other co-authors; C.M.A.P. supervised and coordinated the project.

## Additional information

**Competing interests:** The authors declare no competing financial interests.

