## [Peer Review File · Nature Communications]

Reviewers' comments:

Reviewer #1 (Remarks to the Author):

The authors record single units from rat perirhinal cortex during performance of a task involving a rewarded visually cued turn on a figure of 8 maze. They find neurons with firing fields extending over large parts of the maze.

General comments.

The results are interesting in that functional descriptions of neural firing in perirhinal cortex are rare, beyond some work on object recognition. However, it is not clear exactly what the function of these neurons are. How do they relate to trajectory encoding within the hippocampus (Wood, Dudchencko et al., 2000; Frank, Brown, Wilson, 2000) or the parietal cortex (Nitz, 2006), or to spatial coding by place cells with large spatial firing fields (Jung, Weiner, McNaughton, 1994; Kjelstrup, Solstad et al, 2008)?

Specific comments.

They mention that rats had the option of turning round and choosing the other arm, up to the point of no return (PNR). But I can't see any account of how this was incorporated into the analysis. Were these trials included? Were they compared with trials where the rats did not change their decision? This may be what they are referring to in Figure 4L and the last paragraph of the "Spatial Domain" section, but it's not clear

Is there any way to ask whether the responses are driven more significantly by time or space? If the cell responses appear to be primarily spatial (e.g. left arm v right arm), then why analyse the data in the temporal domain at all?

Is there any way to assess whether these firing patterns are linked to the presence of reward? The discriminatory firing appears to be sustained when they return to the middle arm, where they receive additional reward, and basically reflects the presence of reward in each arm. Were there any 'probe' trials where reward was not administered, for example?

Abstract and Introduction: I find this 'larger task context' and 'large scale knowledge' stuff a bit irritating and misleading. They state that "While [neural coding of an animal's local position, heading direction and velocity of movement] ... are needed to mediate detailed navigation, it is equally useful for animals to encode knowledge of their global position in large-scale environments in which tasks are performed" - that makes no sense to me. I think it's well established by now that more ventral grid cells and place cells could easily code for 'global positions' in the kind of 'large scale environments' they examine (which are 1.1m square!). I don't even know what 'global knowledge' really means.

Figure 1D-F: These panels are not referred to at all in the text, as far as I can tell

Figure 2A-D: It is not clear to me why there are ~3x more 'left' (orange) than 'right'

(purple) trials here? According to the Methods, there are an average of 60 trials in each block (why was this number varied across blocks?), suggesting that this data is collapsed over more than one block - is that the case, with trials arbitrarily split into left and right? Or is this data from more than one block, and if so, did the animals get blocks of left and right turn trials? This information is not given in the Methods (see below). Finally, the threshold of $p < 0.01$ seems high considering the multiple comparisons issues (i.e. the test is performed independently at each time bin, and there are $60 \times 0.25s$ time bins in each 15s plot).

Figure 2E: It is also clear to me why the bars under Fig 2E, showing the significance of rate differences between arms, are thinner on the right hand panel; and significance level at which this test was made is not given in the caption. Are we to assume it was $p < 0.01$ again and, if so, this is quite high considering the multiple comparisons being made ($72 \times 0.25s$ bins in 18s). Finally, can the authors provide any explanation for the fact that the firing rate difference between arms in PRH cells continues to be significant for $>8s$ after the animal has returned to the central waiting area (PRM)?

Figure 3: Firstly, the authors should mark the level of significance on each panel with a dashed line. Secondly, the caption asks us to "note the prolonged nature of discriminatory firing in PRH when the rats traversed the side arms" but, given that the stated threshold for significance at $p < 0.05$ is 1.63, the period of significantly discriminatory firing in PRH (at this very liberal threshold) is $\sim 2s$ - how does this correspond to the $7 \pm 0.3s$ quoted in the text at the bottom of page 4? Furthermore, the average length of time taken for rats to run between PNR and PRM is not given anywhere in the manuscript, but this would greatly facilitate interpretation of the time courses of discrimination shown in this figure (and those in Figure 2)

Figure 4E, F: It might be clearer if mean normalised rates across all units for the preferred and non-preferred arm were plotted here - it is difficult to visualise the decrease in firing rates on the non-preferred arm in PRH, but not CA1, cells here

Methods: Were there an equal number of left and right trials in each session, and was the order randomised? This information does not appear to be provided

Reviewer #2 (Remarks to the Author):

Bos et al. examine the spatial firing properties of perirhinal neurons in the rat during performance of a left/right choice task on a figure-8 shaped track. Choice is determined by the presence of visual target cue to the animal's left or right as it approaches an intersection. The main finding is that a large number of perirhinal neurons exhibit firing over much of one side of the environment. The result is contrasted with the firing of sensory cortical neurons and hippocampal neurons. The latter have place-specific firing that maps the environment with much finer spatial resolution. The main effect appears to be fairly strong and the analyses are largely appropriate. The finding is novel, pointing to a

potentially unique role for perirhinal cortex in encoding of spatial information. The work also stands as one of very few to define strong firing correlates for perirhinal neurons. Finally, the spatial selectivity of perirhinal neurons was related to the amplitude of the hippocampal theta rhythm and perirhinal neurons have theta-modulated firing. These findings reveal a means by which perirhinal cortex may be operating as part of the larger system of brain regions involved in encoding of position and in navigation. My detailed comments follow.

1) The authors use the term "topographical orientation" in the introduction, but do not define what they mean by it. The connection between 'chunks' and 'topography' is unclear. Further, the authors later introduce the term "configural" and "unitizing" to describe potential functions of perirhinal cortex. I would suggest that the authors work hard to define terms more clearly, perhaps cutting them down in number, and to perhaps consider real-world analogies that better complement the task that is actually used. Further, it would seem that the authors could introduce the concept of spatial mapping at different scales.

2) The authors wish to describe the lower rates on the non-preferred arm as decrements or suppressions in firing. Can this really be claimed and is it necessary to do so? The problem is that the authors compare firing to a baseline value and it is unclear what an appropriate baseline is.

3) In the discussion, the authors liken the observed spatial firing properties to the initial description of medial entorhinal cortex grid-patterned firing. This is unnecessary and will invite valid criticisms since the novelty of the perirhinal form of spatial firing does not match that of grid cells.

4) In linearizing the environment, the authors apply a rather low-resolution binning procedure. Does this have to do with the chosen analytical methods and the relatively low firing rates of neurons?

5) The authors present control data to suggest that the firing over one side of the track is likely spatial and not dependent on single responses to sensory stimuli. Why then are so many of the analyses done with respect to time? The back and forth between analysis of firing in the temporal versus spatial domain can be confusing, doesn't appear to accomplish anything, and demands much of the reader. The manuscript would be better off with a simplified set of analyses in the spatial domain. As it is, the peri-event time-based analyses sometimes produce the false sense that L/R discrimination peaks and falls relative to specific times (figure 3). Is it not the case that the variability in distance travelled since the event is causing the effect?

6) The main finding could be interpreted as evidence for working memory in the larger hippocampal system. The authors might do well to raise this topic in discussion and to make reference/contrast to work by the Jung laboratory.

Reply to reviewers' comments on “Firing patterns of perirhinal neurons are sustained across chunks of task environment” by J. Bos et al., submitted to Nature Communications.

Reviewer #1:

1.0. The authors record single units from rat perirhinal cortex during performance of a task involving a rewarded visually cued turn on a figure of 8 maze. They find neurons with firing fields extending over large parts of the maze.

General comments.

The results are interesting in that functional descriptions of neural firing in perirhinal cortex are rare, beyond some work on object recognition. However, it is not clear exactly what the function of these neurons are. How do they relate to trajectory encoding within the hippocampus (Wood, Dudchenko et al., 2000; Frank, Brown, Wilson, 2000) or the parietal cortex (Nitz, 2006), or to spatial coding by place cells with large spatial firing fields (Jung, Weiner, McNaughton, 1994; Kjelstrup, Solstad et al, 2008)?

>> We are glad the reviewer appreciates our findings on firing correlates of perirhinal (PRH) neurons and their description in the paper. The question of how the function of these neurons relates to (dorsal or ventral) hippocampal or parietal coding is indeed an intriguing one. Whereas we recorded from four areas simultaneously, focusing on dorsal hippocampus, PRH and sensory cortices, comparisons to ventral hippocampus and parietal cortex can be made only indirectly, using the existing literature. Based on this approach, and assuming a graded continuous scale of place field sizes between dorsal and ventral hippocampus (cf. Kjelstrup et al. 2008), it is interesting to note that the boundaries of PRH firing fields are primarily defined by spatial decision points (e.g. fig.3, Revision) on the figure-8 maze, whereas our dorsal CA1 cells (recorded simultaneously) show much smaller firing fields, not bound to these points. This suggests that on- and offsets of PRH firing-fields are more tightly bound to bifurcation points than those of hippocampus. However, for ventral hippocampus and parietal cortex this difference with PRH cells remains to be verified.

We amended the Discussion on this important point as follows:

“Although PRH firing behavior could only be compared directly to three brain areas including dorsal area CA1, it is relevant to relate it to other regions. Ventral CA1 neurons express large place fields, which appear as scaled-up versions of fields found in dorsal CA1, and thus likely differ from PRH neurons in that they are scattered across the rat's environment, not showing a demarcation by spatial decision points or other behaviorally significant landmarks^{9,51}. The Left-right selectivity of PRH neurons is also somewhat reminiscent of rat parietal neurons. However, parietal firing patterns were shown to map epochs of route traversals, depending

on behaviors such as turning or forward locomotion⁵², whereas the sustained patterns in PRH could not be accounted for by such specific behaviors (cf. fig. 3L). Regarding the subtle but significant retention effect in PRH observed after the rat's return to the middle lane (fig. 2E, fig. 3, fig. 4 and fig. S2I-L), this type of retrospective coding is similar to that found in CA1 and entorhinal cortex neurons, at least under some task conditions such as spatial alternation⁵³. However, we did not observe retrospective coding in area CA1 of rats performing the visual discrimination task. Also prospective coding of future paths has been reported in area CA1 of rats performing spatial alternation^{53,54}, but this was not found in PRH or CA1 during our visual discrimination task (fig. 2E, fig.3, fig. 4 and fig. S2I-L)."

1.1. Specific comments.

They mention that rats had the option of turning round and choosing the other arm, up to the point of no return (PNR). But I can't see any account of how this was incorporated into the analysis. Were these trials included? Were they compared with trials where the rats did not change their decision? This may be what they are referring to in Figure 4L and the last paragraph of the "Spatial Domain" section, but it's not clear.

>> *These trials were included. However, it occurred very rarely (on average < 1 per session) that a rat fully started walking into one arm and - while in being this arm - changed direction and turned around. Up until the Point of No Return (PNR) the rat was allowed to choose his own route. For a trial to be included in the analyses in our paper, the animal needed to break the Point of No Return beam within 15 seconds after stimulus onset. Given the very low number of trials in which the animal changed running direction while in the initial (i.e., sandpaper-covered) part of the side arms (<1 per session), the rationale was lacking to analyze them in detail. We felt it made little sense to exclude these trials, as the deviant behavior was rare and failed to affect the main results (see e.g. fig. 4 in Original, now renumbered as fig. 3). We now noted in the text that turning on the maze segments before passing the PNR was quite rare (see Methods).*

To clarify the Methods description on this point, we changed the text as follows:

"Both correct (rewarded) and incorrect (non-rewarded) behavioral trials were included in the analyses described here and below, unless noted otherwise. Trials in which the rat would turn around before reaching the PNR and choose the other arm were rare (on average < 1 per session) and were also included, unless noted otherwise."

Figure 4L (as numbered in the Original; fig. 3L in the Revision) is referring to a different situation. After the rat visited the reward site he needed to go back towards the central arm to start a new trial. This happens by forward locomotion as denoted on the x-axis (either in the Left or Right arm). Before entering the central arm, the animal frequently turned around and walked back towards the Point of No Return. This allowed us to do the control analysis shown in Figure 3L. Here we show a correlation of firing rates during movements either in the

correct, forward direction or when the animal was walking back towards the PNR. The positive correlation in Figure 3L shows that if the neuron showed a higher firing rate for the Left arm of the maze compared to the Right arm, while walking forward, this same unit still showed a higher firing rate for the Left compared to the Right side when the animal walked in the opposite direction. To clarify that the analysis in Fig. 3L only included data from the chosen arm segments visited after the rat passed the PNR, we changed the text in the Results section as follows:

"To assess which of these possibilities applied to PRH responses, we contrasted trials in which the animal first passed the PNR, collected a reward and subsequently moved forward or travelled in the opposite direction on the same individual arm, up to the PNR but no further. Here, runs in the opposite direction represent a deviation from the regular task sequence."

Moreover, a possible source of confusion was removed from the Discussion section:

"Furthermore, the strong correlation in Left-Right selectivity between forward and opposite-direction runs occurring after the rat had passed the PNR (Fig. 3L) speaks against a predominant influence of head and eye orientation..."

1.2. Is there any way to ask whether the responses are driven more significantly by time or space? If the cell responses appear to be primarily spatial (e.g. left arm v right arm), then why analyse the data in the temporal domain at all?

>> This point is well-taken. Both the temporal and spatial analyses support that the responses are driven by space. However, the temporal domain allows us to analyze and visualize the data in a temporally precise way (because the PETHs can be synchronized on the rat passing photobeams at the Point of No Return and Point of Return to Middle). For the spatial analyses we used rat positions reconstructed from LED lights on the headstage of the animal (see Methods). These positions were sampled at 25 Hz, video frame rate. This will lead to some small measurement errors, which slightly dilute the on- and offset edges of the firing fields in space (see fig. 3 and 4 in Revision). In the temporal domain we can use photobeam breaks sampled at 32 kHz, which give a better indication of when exactly specific locations were passed. Moreover, the spatial plots (e.g. fig. 3) present a "temporally lumped" picture of the data and thus cannot present data on the high consistency of Left-Right selective responses over trials (see e.g. fig. 2). Thus, PETH-analysis turned out to be an important tool to show the trial-to-trial consistency of unit responses with high temporal precision. Having said this, we recognize the reviewer's hint that (given space as driving factor for the firing correlates) it would be appropriate to pay more attention to spatial analyses, especially once, in the paper, the spatial correlates have been revealed in detail (fig. 3). Thus, we have limited the time-domain analyses to fig. 1 and fig. 2, and converted the original fig. 5 (now fig.4) from the temporal into the spatial domain. Furthermore, we

moved the figure on Discrimination Score as a function of time (fig. 3 as numbered in the Original) to the SI section (new fig. number S3).

We also made the following changes in the text to clarify the use and precision of analyses in the temporal domain:

- 1. In the Methods section we now explicitly noted: "Two of these photobeams were positioned to record the rat passing the PNR and PRM".*
- 2. In the introductory paragraph of the Results we added: "This passage was recorded by a photobeam break, and similarly we recorded the moment when rats passed the point of returning to the middle lane (Point of Returning to Middle, PRM)".*
- 3. The Results section 'Sustained perirhinal firing patterns' was amended with: "Below we will first present analyses of PRH firing behavior in the time domain, because our behavioral setup allowed us to accurately correlate firing-rate changes to the passage of the PNR and PRM in time. Moreover, the repetitive nature of the task permitted us to study the consistency of firing correlates over trials. Subsequently, analyses in the spatial domain will be presented."*

1.3. *Is there any way to assess whether these firing patterns are linked to the presence of reward? The discriminatory firing appears to be sustained when they return to the middle arm, where they receive additional reward, and basically reflects the presence of reward in each arm. Were there any 'probe' trials where reward was not administered, for example?*

>> In Figure 4 (Revision) we showed several control analyses. Fig. 4A contrasts correct (rewarded) versus incorrect (non-rewarded) trials. Let us recall that, in incorrect trials, reward was administered neither in the side arms nor in the middle arm. In color we show the contrast within one and the same arm (i.e., the orange line in 4A shows correct – incorrect responses in the preferred arm - the arm with the highest average firing rate- and the purple line shows the same contrast for the non-preferred arm). We found no differences in firing rate when the animal received a reward versus when he did not - neither at the preferred nor non-preferred side. However, when we look at differences between the Left and Right side - quantified for either rewarded (correct; black line) or unrewarded (incorrect; grey line) trials, we observed large sustained differences between sides. These results clearly indicate that the Left-Right response differences were not reward-driven.

As concerns sustained firing when rats returned to the middle arm, indeed a residual discrimination by firing rate (or Normalized Rate Difference, fig. 2E) was observed when the rat returned to this arm. The sustained middle-arm firing is now further illustrated in Supplementary fig. S2, also showing behavioral-path controls. The cause of this effect is unknown but, given the above results, is unlikely to relate to the reward which is applied there in every correct trial (i.e., this reward is not a discriminating factor between Left and Right trials). The steep drop of sustained firing-rate differences at the point of return to middle (e.g. fig.2E, fig. 3D)remains the main effect during the return phase of the trial.

We adjusted the text to describe the residual arm effect, and its interpretation as follows:

- In the Results section: "Discriminatory firing by PRH neurons strongly increased around the passage of the PNR and steeply decreased when passing the PRM. In addition to this steep decrease, a residual significant rate-difference remained present when the rat had returned to the middle lane, an effect that was not observed in area CA1 or S1BF (fig.2E)."

- In the Discussion: "Because reward was administered only in correctly performed trials, discriminatory firing is unlikely to be explained by differences in its delivery or expectancy. In addition to the strong discrimination PRH neurons made between the Left and Right arm, a more subtle residual Left-Right discrimination was observed after rats returned to the middle arm (fig. 2E; fig. 3D; Supplementary figure S2I-L). The exact cause of this effect is unknown, but is unlikely related to rewards applied in the middle arm because it remained visible when the Left versus Right arm trials were contrasted given correct or incorrect performance (fig. 4A). In other words, even when no middle-arm reward was provided, a Left-Right difference was observed. Our data further suggest that residual discriminatory firing, occurring when rats had returned to the middle lane, is an effect of trial history and reflects retrospective coding (i.e., a dependence of PRH firing on where the rat came from in the foregoing trial). Although it is notable that this retrospective activity occurred under different behavioral conditions (fig. 4A-D), it remains to be examined what the main factors are determining the strength of this effect."

1.4. Abstract and Introduction: I find this 'larger task context' and 'large scale knowledge' stuff a bit irritating and misleading. They state that "While [neural coding of an animal's local position, heading direction and velocity of movement] ... are needed to mediate detailed navigation, it is equally useful for animals to encode knowledge of their global position in large-scale environments in which tasks are performed" - that makes no sense to me. I think it's well established by now that more ventral grid cells and place cells could easily code for 'global positions' in the kind of 'large scale environments' they examine (which are 1.1m square!). I don't even know what 'global knowledge' really means.

>> Our use of terms like 'larger task context' and 'large scale knowledge' was inspired by the literature on topographic orientation disorders in humans with parahippocampal lesions. For instance, Maguire (1997; as cited) defines topographical orientation as "the ability of humans to orient and navigated successfully in the large-scale spatially extended environments that constitute the real world". These patients have problems getting lost while navigating in unfamiliar or even familiar environments (Maguire 1997; Aguirre and D'Esposito 1999; Bird et al. 2010, as cited). Because of their topographic orientation difficulties, suggesting impaired large-scale navigation abilities, we feel the correspondence with the current rodent findings on PRH is rather striking. But we agree with the reviewer that this correspondence does not mean that PRH would be the (only) brain area mediating

large-scale spatial navigation, and certainly ventral entorhinal grid cells and ventral hippocampal place cells should not be left out of the equation. However, what remains as a very interesting difference between PRH cells and CA1 place cells, is not only the difference in field size (e.g. fig. 3 in Revision; but of course related to the dorsal CA1 recording sites) but also that the CA1 correlates are scattered across all parts of the maze whereas the spatially sustained nature of PRH firing pattern is bounded by the spatial decision or branch points (see fig. 2, 3) along the way. Metaphorically, one may compare this to a situation of a car driver navigating through a city, where he needs to get from neighborhood A to a house in a remote sector B, and where the macrogeometrical knowledge of T-junctions and intersections is needed in the first place to map a route for navigation. Only when getting close to B, more detailed knowledge (e.g. of which houses he will pass in succession) will be needed.

In the Introduction we removed references to “global position”, because it is more widely accepted to refer to spatial scales (see e.g. Kjelstrup et al. 2008, as cited):

“...it is equally useful for animals to encode macrogeometrical knowledge to navigate in large-scale environments to reach a distant goal.”

We then continue to define “topographical orientation” more precisely, following Maguire (1997):

“The ability to orient and navigate successfully in large-scale environments is also referred to as ‘topographical orientation’⁶.”

To introduce this ability more clearly to readers unfamiliar with this clinical topic, we add:

“This capacity can be illustrated by a car driver who navigates through a city to get from district A to a house in a remote quarter B. Given this task, he first needs to apply large-scale knowledge of e.g. T-junctions and neighborhoods to map an overall route for navigation. Only when getting close to B does he require more detailed knowledge, e.g. of which houses he will pass in succession. Navigation through large-scale environments may be facilitated by ventral hippocampal and entorhinal cell populations, showing larger scales of spatial coding than their dorsal counterparts^{1,9}. However, both dorsal and ventral hippocampal cells have place fields scattered across the environment, and it remains unknown whether and how “chunks” of environments or spatial trajectories, as demarcated by decision points, are coded.”

Further changes were made to the Discussion, see point 1.0 above.

1.5. Figure 1D-F: These panels are not referred to at all in the text, as far as I can tell

>> Thank you for noting this. We have now referred to them in the revised text.

1.6. Figure 2A-D: It is not clear to me why there are ~3x more 'left' (orange) than 'right' (purple) trials here? According to the Methods, there are an average of 60 trials in each block (why was this number varied across blocks?), suggesting that this data is collapsed over more than one block - is that the case, with trials arbitrarily split into left and right? Or

is this data from more than one block, and if so, did the animals get blocks of left and right turn trials? This information is not given in the Methods (see below). Finally, the threshold of $p < 0.01$ seems high considering the multiple comparisons issues (i.e. the test is performed independently at each time bin, and there are 60 x 0.25 s time bins in each 15 s plot).

>> Using pseudo-random sequences we made sure that of every 10 trials, five had the CS+ presented on the left and five had the CS+ on the right side. A block consisted of 10 trials, but actually the concept of 'block' is not required to explain the session structure and was therefore left out of the Revision. Throughout the session no more than 3 same-side trials were presented in succession.

Indeed, the animals performed on average 60 trials per session.

Figure 2A-D shows single-unit responses from one session. Having more Left than Right side responses means that the animal chose to go left more often than right, thus making more incorrect responses on the left side. Thus, because of erroneous choices, there can be a difference between the number of CS+ presentations on the left or right, and the number of behavioral choices made to each side. Furthermore, we should stress that correct/incorrect performance did not matter for the specific Left-right discriminations coded by PRH cells (see fig. 4, Revision).

We explained the matter in the Methods as follows:

"Using pseudo-random sequences we made sure that, for each series of 10 consecutive trials, five had the CS+ presented on the Left and five had the CS+ on the Right side. Throughout the session no more than three same-side trials were applied in succession."

To correct for multiple comparisons we compared each test bin (i.e. the bin of which the neural response is tested for statistical significance) to eight successive baseline bins (2s total duration). The baseline was taken from the last seconds of the inter-trial interval. A given test bin was only designated as significant when the bin was significantly different from ALL eight baseline bins at $p < 0.01$. Thus, the $P < 0.01$ criterion had to be met eight times in succession (corresponding to 8 times a logical AND operation; see also Van Duuren et al. 2007; Lansink et al. 2009, as cited) which in general is quite conservative as compared to some other methods (e.g. Cromwell and Schultz 2003, J. Neurophys. 89: 2823-2838).

We clarified the statistical testing of binned neural responses in the text as follows:

"To compensate for multiple comparisons, each bin of interest was tested against eight, contiguous baseline bins in a late time segment of the inter-trial interval, at -3 to -1 s relative to the auditory sound at trial onset (bin width: 250 ms). To qualify as significant, the P-values of a test bin needed to exceed the significance threshold ($P < 0.05$) against each baseline bin. Thus, the $P < 0.01$ criterion had to be met eight times in succession (cf.^{61,64})."

1.7. Figure 2E: It is also clear to me why the bars under Fig 2E, showing the significance of

rate differences between arms, are thinner on the right hand panel; and significance level at which this test was made is not given in the caption. Are we to assume it was $p < 0.01$ again and, if so, this is quite high considering the multiple comparisons being made (72 x 0.25 s bins in 18 s). Finally, can the authors provide any explanation for the fact that the firing rate difference between arms in PRH cells continues to be significant for > 8 s after the animal has returned to the central waiting area (PRM)?

>> Thank you for noting this error. The thickness of all error bars should have been the same. This has been corrected. Concerning the significance level in fig.2E: see previous point. As before, each test bin needed to meet the $P < 0.01$ threshold against all eight baseline bins in order to be taken as significant. We have inserted the significance level in the caption of 2E:

"Each test bin needed to meet the $P < 0.01$ threshold against eight contiguous baseline bins in order to be designated as significant (see Methods)."

Furthermore, we included a reference to fig. 2 in the methodology section.

As regards the firing-rate difference being significant for > 8 s in the return phase, this point is well taken. As confirmed above, there is indeed some residual differential activity after the animal has passed the Point of Return to Middle (PRM). This activity may result from a hysteresis effect, or in other words a history-dependence defined by whether the animal returned to the middle lane from the Left vs. Right arm. However, given the stark decrease in NRD (normalized rate difference) seen at the Point of Return to Middle ($t = 0$ s in fig.2E, right panel) we do not think this diminishes the overall significance of the Left- Right chunking effect demonstrated in the figure. For changes in the Results and Discussion sections, see point 1.3 above.

1.8. Figure 3: Firstly, the authors should mark the level of significance on each panel with a dashed line. Secondly, the caption asks us to "note the prolonged nature of discriminatory firing in PRH when the rats traversed the side arms" but, given that the stated threshold for significance at $p < 0.05$ is 1.63, the period of significantly discriminatory firing in PRH (at this very liberal threshold) is ~ 2 s - how does this correspond to the 7 ± 0.3 s quoted in the text at the bottom of page 4? Furthermore, the average length of time taken for rats to run between PNR and PRM is not given anywhere in the manuscript, but this would greatly facilitate interpretation of the time courses of discrimination shown in this figure (and those in Figure 2)

We propose to leave this figure out of the new (main) Manuscript, first to put less emphasis on the temporal dimension of analysis (see also points 1.2 and 2.5), and second because, thanks to this comment, we realize that the figure is potentially confusing. In the Original (fig.3 legend), we had already included a comment on the source of the slow decline (A) and slow rise of the PRH Discrimination Score (B), but some more explanation is appropriate. Due

to the nature of time-locked representations and given the behavioral variability in the speed it takes a rat to complete a trial, it is logical that one will observe a decline in Discrimination score, the further each time point of observation gets away from $t=0$. This is because some trials are completed rapidly (so that Left or Right arms travels are completed in only few seconds) and others take longer to complete; this behavioral scattering or 'jitter' across time causes a dilution effect and decline at times after $t=0$ s, as the Discrimination Score rapidly drops toward zero once the quick arm runs have been completed (a similar, albeit less steep decline is visible in the NRD time course of fig. 2E). Furthermore we note that the Discrimination score is computed as the T-statistic $T_a = (D_a - D_s)/sd$ (see Methods), clarifying that this measure will decrease when the standard deviation (sd) of the distribution of D_s (shuffled scores of firing-rate differences) increases. This occurs particularly in the case of behavioral variability affecting whether or not the rat is located in the critical Left-Right arm portions, and probably also because of other sources of behavioral variability affecting the strength of the Left-Right selectivity (e.g. due to stopping, grooming, reward consumption). Thus we have moved fig.3 out of the new version of the main manuscript and placed it in the Supplementary section (now fig. S3). As requested by the reviewer, we have now mentioned the average length of time taken for rats to run between PNR and PRM in the legend of fig. S3, together with the length of the average time periods during with the Discrimination score was significantly elevated in each structure (we applied a different, improved calculation of these durations than used in the Original, taking into account the variable passage times per trial).

The legend of Fig. S3 (formerly fig.3) was amended as follows:

" The average duration of the time segments showing a significant Left-Right Discrimination Score was $1,96 \pm 0.10$ s (S1BF), $2,43 \pm 0.11$ s (CA1), $4,11 \pm 0.17$ s (PRH) and $2,51 \pm 0.32$ s (V1M; these durations cannot be estimated from the graphs because they are based on individual trial segments). The duration for PRH cells was significantly higher than for the other three areas ($P < 0.05$). The average time it took rats to pass from the PNR to PRM points was 8.83 ± 0.36 s. That Discrimination Scores were elevated for shorter time periods than these behavioral run periods is explained by the nature of time-locked representations and given the behavioral variability in the speed it takes a rat to complete a trial. In other words, because some trials are completed rapidly (within a few seconds) and others take longer to complete, a behavioral scattering or 'jitter' occurs relative to $t=0$ s, causing a dilution effect and thus a decline of the Discrimination score in (A) and a slow rise in (B). Furthermore, the Discrimination score is computed as the T-statistic involving the standard deviation (see Methods), clarifying that this measure will decrease when the standard deviation of the distribution of shuffled firing-rate differences increases, which occurs because of behavioral variability."

And, concluding:

"Overall, this figure illustrates that PRH cells show significantly more sustained firing than cells in the other three areas, but also that it is more appropriate to quantify these firing patterns as a function of space rather than time."

1.9. Figure 4E,F: It might be clearer if mean normalized rates across all units for the preferred and non-preferred arm were plotted here - it is difficult to visualize the decrease in firing rates on the non-preferred arm in PRH, but not CA1, cells here

We appreciate this valuable suggestion. We have now added the mean normalized rates for the non-preferred and preferred arm and the Normalized Rate Difference (NRD), for both PRH and CA1 (Revision: fig.3, D-I). Whereas the difference between PRH and CA1 cells for the non-preferred arm is now more clearly visible, the similarity between PRH and CA in the graphs for the preferred arm (fig. 3E vs. 3H, respectively) may create the (false) impression that there is little difference (N.B., this impression arises because many 'punctate', high-peak place fields of CA1 add up in the mean, as the preferred arm is defined, in case of CA1 cells, by neurons expressing clear place fields in that arm). Emphasizing the differences in firing-fields between individual PRH vs. CA1 cells (see also fig. 3J), we averted this risk, first by inserting the following text in the legend of fig. 3C and D:

"The mean Normalized Rate Difference is plotted on top of the color-coded survey of all units, with the baseline value plotted as a dashed line and spatial ranges significantly deviating from baseline indicated by the horizontal lines below. Vertical colored lines correspond to the spatial decision point (cyan), PNR (blue), reward site (purple) and PRM (pink; see also C)."

Furthermore, the legend of fig. 4G-I was amended as follows:

"Note that the mean NRD panel for CA1 cells has a spatially extended enhancement due to the fact that CA1 cells expressing a place field for the Preferred arm contribute 'punctate' responses adding up in the mean NRD. No such enhancement is seen in the CA1 Non-preferred arm plots because, by definition, this arm is devoid of strong place fields."

1.10. Methods: Were there an equal number of left and right trials in each session, and was the order randomized? This information does not appear to be provided

>> Yes, there were an equal number of left and right trials in each session, at least if a left trial is taken to mean that the CS+ was presented on the left. See above, point 1.6. Trials were selected pseudorandomly such that every 10 trials contained 5 trials with the CS+ on the Left and 5 with CS+ on the Right, and there were no more than 3 trials to the same side in succession. We amended the Methods on this point.

Reviewer #2 (Remarks to the Author):

2.0. Bos et al. examine the spatial firing properties of perirhinal neurons in the rat during performance of a left/right choice task on a figure-8 shaped track. Choice is determined by the presence of visual target cue to the animal's left or right as it approaches an intersection. The main finding is that a large number of perirhinal neurons exhibit firing over much of one side of the environment. The result is contrasted with the firing of sensory cortical neurons and hippocampal neurons. The latter have place-specific firing that maps the environment with much finer spatial resolution. The main effect appears to be fairly strong and the analyses are largely appropriate. The finding is novel, pointing to a potentially unique role for perirhinal cortex in encoding of spatial information. The work also stands as one of very few to define strong firing correlates for perirhinal neurons. Finally, the spatial selectivity of perirhinal neurons was related to the amplitude of the hippocampal theta rhythm and perirhinal neurons have theta-modulated firing. These findings reveal a means by which perirhinal cortex may be operating as part of the larger system of brain regions involved in encoding of position and in navigation. My detailed comments follow.

>> We would like to thank this reviewer as well, both for his/her constructive comments and positive remarks on the novelty and significance of the findings.

2.1. The authors use the term "topographical orientation" in the introduction, but do not define what they mean by it. The connection between 'chunks' and 'topography' is unclear. Further, the authors later introduce the term "configural" and "unitizing" to describe potential functions of perirhinal cortex. I would suggest that the authors work hard to define terms more clearly, perhaps cutting them down in number, and to perhaps consider real-world analogies that better complement the task that is actually used. Further, it would seem that the authors could introduce the concept of spatial mapping at different scales.

>> In the Introduction of the Revision, we have clarified the term "topographical orientation" in accordance with the clinical literature on this topic, see point 1.4 above. Furthermore, in the PRH literature, terms like "unitizing" and "configural" have both been used, but we agree that the meaning of "configural" is not always clear in this context. Unitization is a concept from psychology and is defined as the process by which two or more previously separate items or stimulus components become represented as a single unit (see e.g. Graf and Schacter 1989, as cited). For instance, features of a face become integrated into a single face-object, and parts of a compound word come to form a new word by association (cf. Sauvage et al. 2008; Nat Neurosci. 11: 16-18). In the new version we therefore largely avoid "configural" but maintain the term "unitizing". Furthermore, we now included an explanation of the term "unitizing" in the Introduction:

"Such a "unitizing" function of the PRH, serving to integrate two or more previously separate items into a single representation²⁹, is supported by lesion studies^{23,30,31} ... "

Finally, we have introduced a real-world analogy (navigation by car drivers in a city) and introduced mapping at different scales, along with chunking of trajectory segments by spatial decision points, in the Introduction (see point 1.4 above).

2.2. The authors wish to describe the lower rates on the non-preferred arm as decrements or suppressions in firing. Can this really be claimed and is it necessary to do so? The problem is that the authors compare firing to a baseline value and it is unclear what an appropriate baseline is.

This point is well taken. In the Revision we clarify that there are increments and decrements with respect to the baseline firing of each cell, which we have defined as its firing rate in the ITI (spent in the central arm). Furthermore, we clarify that taking the central arm to compute a baseline is essentially arbitrary, although in agreement with most of the literature taking ITI (intertrial interval) firing rates as baseline.

Regardless of the choice of baseline, it does remain the case that single PRH units can increase their firing rate in one side of the maze and decrease it in the other. As we noted in the Original, some cells have firing rates significantly discriminating both the Left and Right arm from the Middle lane (baseline) and with different signs of these changes (up vs. down; e.g. fig. 2B). It appears worth clarifying that PRH cells are thus capable not only of making a binary spatial distinction, but also graded distinctions between multiple chunks of the maze. Therefore we amended the Methods as follows:

"Although this interval is commonly used in the electrophysiological literature to compute a baseline, it should be emphasized that, in view of the type of spatial selectivity of PRH cells, this choice of baseline is essentially arbitrary (cf. fig. 2B)."

In the Results section we added, first:

"firing-rate changes are denoted as increments and decrements relative to the baseline, computed during the intertrial interval, spent in the Middle lane".

Furthermore, we removed the term "suppressive component" and emphasized the relative nature of the changes:

"The joint expression of sustained increments and decrements by the same cells suggests that PRH firing patterns contain an activating and a deactivating component, as defined relative to baseline, both of which contribute to strong discrimination."

To apply more caution in the Discussion section, we now state:

"This deactivation may correlate either with a loss of excitation or with a powerful feedforward inhibition",

which is followed up by the following conclusion to this paragraph:

"However, the precise mechanisms underlying discrimination between maze segments awaits further investigation. Here, the main point worth emphasizing is that PRH cells are capable not only of making a binary spatial distinction, but also of making graded

distinctions between multiple maze segments (e.g. Left versus Right versus Middle, e.g. fig. 2B)."

2.3. In the discussion, the authors liken the observed spatial firing properties to the initial description of medial entorhinal cortex grid-patterned firing. This is unnecessary and will invite valid criticisms since the novelty of the perirhinal form of spatial firing does not match that of grid cells.

>> Thank you for this comment. We have deleted this statement.

2.4. In linearizing the environment, the authors apply a rather low-resolution binning procedure. Does this have to do with the chosen analytical methods and the relatively low firing rates of neurons?

>> Indeed this choice relates to the relatively low firing rates and analytical methods, but also to the rat's behavior. We used a bin size of ~ 9 cm (see Methods; along the running direction on the maze), which has been used more often in the literature (e.g. 10 cm, applied in Davidson et al. 2009, Neuron 63: 497-507). We have studied the effect of reducing spatial bin size, but found that we did not have enough coverage of all of these smaller bins (i.e., the spike count per bin becomes too low for accurate firing-rate estimates for some parts of the maze). Especially in the initial parts of the side arms, close to the decision point (fig. 1A), the running speed of the animal is usually high. Additionally, the headstage LEDs, which are used to track the animal's position, are fairly high up. Head movements, e.g. from side to side or from downwards to upwards, cause some spatial jitter, to which larger spatial binning is less sensitive than smaller binning.

2.5. The authors present control data to suggest that the firing over one side of the track is likely spatial and not dependent on single responses to sensory stimuli. Why then are so many of the analyses done with respect to time? The back and forth between analysis of firing in the temporal versus spatial domain can be confusing, doesn't appear to accomplish anything, and demands much of the reader. The manuscript would be better off with a simplified set of analyses in the spatial domain. As it is, the peri-event time-based analyses sometimes produce the false sense that L/R discrimination peaks and falls relative to specific times (figure 3). Is it not the case that the variability in distance travelled since the event is causing the effect?

>> See point 1.2, where we motivate our analyses in the time domain with two reasons (1. trial-to-trial reproducibility, and 2. precision of firing changes relative to task events). However, we do understand the reviewer's concern and we agree that L/R discrimination peaks and falls create a false sense of being related to specific times (see our reply to Reviewer 1 on Figure S3, Revision). Therefore we now bring more consistency to the order of

the analyses (avoiding back and forth) by confining the time-domain analyses to Fig. 1 and Fig. 2, and then continuing in the spatial domain (except for Fig. 6, set in the frequency domain). Thus, Fig.3 has now been moved to the Supplementary Information (as fig. S3), and Fig. 5 was adapted to display the control analyses (for stimuli, behavioral parameters etc.) in the spatial domain.

2.6. The main finding could be interpreted as evidence for working memory in the larger hippocampal system. The authors might do well to raise this topic in discussion and to make reference/contrast to work by the Jung laboratory.

>> We have raised this topic in the Discussion now (referring to Baeg et al. 2003) and propose several arguments against this hypothesis. First, whether or not a trial included a memory component did not matter for PRH discriminatory firing patterns (fig. 4, Revision). Second, a correlate of working memory would be expected to be in place until the relevant action and reward acquisition have occurred, but not after these events. In contrast, we found that the sustained firing patterns continue well after visiting the reward sites, declining when the rat returns to the middle lane (see fig.1-4). Third, after passing the Point of No Return, no working memory is required to successfully complete the rest of the trial, yet this passage marks the time period where sustained firing is most strongly visible. Finally, even when the rat deviates from the regular trial sequence and walks in the opposite direction on an arm, the spatially selective firing remains basically intact (fig. 3L). We have amended the Discussion on this point as follows:

"The sustained nature of firing brings up the question whether PRH neurons may encode working memory operations, as has been reported for instance in rat medial prefrontal cortex⁵⁵. Several results argue against the (classic) idea of working memory being a main function of the PRH coding we report here. First, whether or not a trial included a memory component did not matter for PRH discriminatory firing (fig. 4D). Second, a correlate of working memory would be expected to be in place until the relevant action and reward acquisition have occurred, but not afterwards. In contrast, we found that the sustained firing patterns continued well after the reward sites had been visited, declining when the rat returned to the middle lane. Third, after passing the PNR, no working memory was required to successfully complete the rest of the trial, yet the PNR-to-PRM trajectory was marked by a pronounced sustained firing. Finally, even when the rat deviated from the regular trial sequence and walked in the opposite direction on an arm, spatially selective firing remained intact (fig. 3L). However, in a more general sense the PRH code may serve to keep track of the animal being localized on a large maze segment during task performance."

Reviewers' comments:

Reviewer #1 (Remarks to the Author):

The authors have answered many of my questions, but a major issue remains. The paper reports neuronal firing in perirhinal cortex characterised as “sustained across chunks of task environment.” However, they may just be large “place fields” as seen in ventral hippocampus, which would change the interpretation somewhat away from the focus on task-chunking. In reply to this point, they note that their firing patterns were demarcated by spatial decision points, whereas place fields tend to be scattered across the environment. As this is an important distinction, this difference needs to be quantified – to what extent were increases in firing rate co-located with spatial decision points?

Reviewer #2 (Remarks to the Author):

The authors have responded well to the issues/concerns addressed in my initial review. The associated changes to the manuscript text and those related to other referee responses have, in my opinion, resulted in a much clearer, thorough consideration of the data. I continue to be of the opinion that the work has importance in driving new ideas about the role of the perirhinal cortex in the larger cortico-hippocampal system as it relates to navigation.

I have two remaining concerns:

1) On line 331, the authors make the provocative statement that the findings cast doubt on perirhinal cortex as functioning mainly to encode the “what” of visual information. I find that just a bit premature given that the system may function in more than one way under different circumstances and given that encoding of both types of information (visual items and spatial chunks) is entirely possible. I would suggest the authors add some nuance to the statement here, perhaps allowing that the “chunks” encoded in the present study could conceivably be considered as “objects” of a sort.

2) On line 290, the authors insert a comment concerning the nature of firing activity in the posterior parietal cortex that is quite simply wrong. Numerous studies, including the one cited, have at this point shown the parietal cortex encoding of spatial position to NOT be epiphenomenal to encoding of specific actions. The authors need to reconsider how parietal cortex firing patterns may or may not be qualitatively different than what they observe and where their paradigm does or does not allow for a direct comparison.

Reply to reviewers' comments on "Firing patterns of perirhinal neurons are sustained across chunks of task environment" by J. J. Bos et al., submitted as Revised manuscript to Nature Communications.

Reviewer #1:

The authors have answered many of my questions, but a major issue remains. The paper reports neuronal firing in perirhinal cortex characterized as "sustained across chunks of task environment." However, they may just be large "place fields" as seen in ventral hippocampus, which would change the interpretation somewhat away from the focus on task-chunking. In reply to this point, they note that their firing patterns were demarcated by spatial decision points, whereas place fields tend to be scattered across the environment. As this is an important distinction, this difference needs to be quantified – to what extent were increases in firing rate co-located with spatial decision points?

dat er een fundamenteel verschil is tussen de cellen die wij vinden en de place fields in hpc, omdat er sterke & sustainede negatieve modulaties zijn.

This point is well taken. As noted before, we did not record in ventral hippocampus, but we can compare the distribution of PRH firing activity with dorsal CA1 place fields. As documented in our paper, PRH firing displays different patterns on the figure-8 maze as compared to CA1 firing. CA1 neurons have baseline firing rates very close to zero. As a result, deviations from baseline will be positive; this makes it fairly easy to compute place field boundaries. PRH baseline firing rates, however, are usually non-zero and, as shown in Figures 1-3, PRH units show both in- and decreases in firing rate relative to the middle lane. Units can show both increases and decreases across the different arms of the maze. It should be noted in particular that the strong, sustained negative modulations with respect to a non-zero baseline firing rate are not shown by CA1 cells.

Given the nature of PRH firing patterns, a straightforward way to compare PRH and CA1 activity might be to take the normalized rate difference (NRD) between the preferred and non-preferred arm (Figure 3D and G). These difference scores have a zero baseline with positive modulations. However taking the NRD could artificially emphasize boundaries around the middle bins. Because we are subtracting firing rates from all bins corresponding to trials to one side from trials to the other side, and because both trial types include bins in the middle arm, thus overlapping in space, their firing activity changes would be subtracted out, whereas this does not apply to bins on the side arms. To avoid this problem we have separately calculated firing fields for the preferred and non-preferred arm.

To transform PRH activity (with its decrements and increments) into quantities that can be compared to CA1 place field responses (having zero baseline and coding by increases), we first subtracted baseline activity from the preferred and non-preferred responses. The baseline was determined for each unit as the average firing rate across all middle and side arm bins for both the preferred and the non-preferred side. For the non-preferred side, the rates were multiplied by -1 to turn the negative deflections into positive values. Because of the way the preferred side was defined (side with highest firing rate) we only used CA1 firing fields from this side to compare PRH firing fields with. Only units with a discrimination score of at least 0.5 (see methods) were included in this analysis.

Next we computed firing fields using a method similar to those used to determine hippocampal place fields (e.g., Johnson and Redish J. Neurosci., 2007; Huxter et al. Nature, 2003, as cited in Revision). The data were treated as circular, i.e. the last bin of a side arm, bin 30, was considered to be adjacent to the first bin of the middle arm, bin 1 (the return branch point). Per single unit the data were smoothed using a 5-point Gaussian. Firing fields were determined by taking the largest number of adjacent bins with a firing rate higher than 20% of the maximum rate. For CA1 units a minimum of 3 adjacent bins was required to enter the analysis. A PRH unit was included as having an extended firing field if the number of contiguous bins reaching > 20% was at least nine.

In area CA1, 100% (373/373) of the units were included in the analysis. In the PRH 52.2% (182/349) showed sustained activity changes on the preferred side and 88.0% (307/349) of the units showed these on the non-preferred side. Start and end points of the firing fields were defined relative to running direction and their distributions are shown in Supplementary Figure S5. First we focussed on the question whether the start and end points of CA1 place fields are scattered more across the maze than those of PRH firing fields. Indeed, the circular variance of the CA1 distributions was significantly higher than for PRH (both for start and end points, and both for the preferred and non-preferred arms of PRH cells; Supplementary Fig. S5, panel E; statistics based on confidence intervals).

Next we assessed the alignment of the firing fields to the decision point (exit from middle lane, Bin 9, see Suppl. Figure S5) and the return branch point (start of middle lane, Bin 1) by calculating the spatial offsets between the start and end points of the firing fields and the decision point and return to middle, respectively. The results from this analysis are displayed in Supplementary Fig. S5, panels F and G.

In line with Figure 3E-F, deactivations of PRH cells in the non-preferred arm corresponded more sharply to the spatial decision points than the activations of PRH cells in the preferred arm (Supplementary Fig. S5F), whereas both PRH activations and deactivations clearly showed end points around the return branch point (in contrast to CA1 cells; Supplementary Fig. S5G). The start points for the PRH sustained activities on the preferred side were more variable than those of the non-preferred side. However, even these start points are significantly less scattered across the maze than those of CA1, as shown by the distributions of variance (circular variance and 95% confidence interval in bins; Preferred arm, PRH_{start} : 2.81, 2.45-3.17; PRH_{end} : 2.72, 2.26-3.11; $CA1_{start}$: 3.72, 3.38-4.01; $CA1_{end}$: 3.73, 3.38-4.03. For the Non-Preferred arm, PRH_{start} : 2.77, 2.47-3.03; PRH_{end} : 2.34, 2.01-2.67; see Suppl. Figure S5E).

As a result of these additional analyses, the following changes were made in the paper:

- A paragraph was added to the Methods section, concerning the determination of start and end points of firing fields; this was supplemented with several further specifications of analyses in the spatial domain, added to the preceding Methods text;
- The new analyses and results were explained in the main text and illustrated in a new Supplementary Figure (S5);
- A brief recapitulation of these results was added to the Discussion.

Reviewer #2:

The authors have responded well to the issues/concerns addressed in my initial review. The associated changes to the manuscript text and those related to other referee responses

have, in my opinion, resulted in a much clearer, thorough consideration of the data. I continue to be of the opinion that the work has importance in driving new ideas about the role of the perirhinal cortex in the larger cortico-hippocampal system as it relates to navigation.

I have two remaining concerns:

1) On line 331, the authors make the provocative statement that the findings cast doubt on perirhinal cortex as functioning mainly to encode the “what” of visual information. I find that just a bit premature given that the system may function in more than one way under different circumstances and given that encoding of both types of information (visual items and spatial chunks) is entirely possible. I would suggest the authors add some nuance to the statement here, perhaps allowing that the “chunks” encoded in the present study could conceivably be considered as “objects” of a sort.

Around line 331 we stated: "Because individual task elements, such as auditory cues, are co-represented in PRH with large spatial segments, even by the same cells (Supplementary Fig. S2), the findings argue against a rigid separation of "what" vs. "where" processing in the medial temporal lobe (cf.¹⁵) and favor an integrative function of PRH, combining chunking of maze segments with coding of individual items".

Thus, we did not intend to cast doubt per se on perirhinal cortex as functioning to encode the “what” of visual information, but rather used the results to argue that "what" vs. "where" processing may be less strictly segregated than previously thought, as PRH appears to co-represent both maze chunks and individual items. We still feel that this statement - as an empirical argument - is warranted by the data, but agree that some nuance is in place here. We thus added:

"From a conceptual viewpoint, both "maze chunks" and other items could be argued to be "objects", defined as composites of informational elements bound by common multisensory and spatiotemporal properties⁶¹. Thus, the current findings do not contradict a role of PRH in coding "what" information per se."

2) On line 290, the authors insert a comment concerning the nature of firing activity in the posterior parietal cortex that is quite simply wrong. Numerous studies, including the one cited, have at this point shown the parietal cortex encoding of spatial position to NOT be epiphenomenal to encoding of specific actions. The authors need to reconsider how parietal cortex firing patterns may or may not be qualitatively different than what they observe and where their paradigm does or does not allow for a direct comparison.

We agree with the reviewer that posterior parietal cortex (PPC) firing patterns during freely moving, navigational behavior are more complex than captured by our brief statement on this in our Discussion. We aimed to be concise here, and were led to make this statement based on Nitz' (2006, as cited) observation that "parietal activity patterns were independent of movement direction and spatial position and scaled with changes in the distance between the execution points of succeeding behaviors". Although other authors and studies have also emphasized body-movement correlates of PPC neurons during freely moving behaviors (e.g.

Whitlock et al. 2012; Nitz 2012), we acknowledge the evidence that spatial information (e.g. on the animal's allocentric position in the larger environment) can co-determine PPC firing. Taking into account that only indirect comparisons between PPC and PRH can be made at this point, the combined evidence does suggest that the two structures are different in that the PRH predominantly shows spatially extended fields of firing (current study) whereas PPC shows a higher abundance of transient firing patterns during performance of particular behaviors (e.g. left body turns). This difference seems to be relative as some PPC neurons fire during prolonged stretches on a maze (Whitlock et al. 2012; Nitz 2012). Naturally we agree that a direct comparison by simultaneous PPC and PRH recordings during maze behavior would be required to test this suggestion. In summary, our Discussion of PPC correlates was rephrased as follows:

"A number of studies suggest that, during freely moving behavior, rat posterior parietal cortex predominantly shows - often transient - firing patterns correlated to the animal's progression through a route, including performance of particular behaviors (e.g. body turns) and movement direction (e.g.^{52,53}) whereas PRH coding is marked by more sustained firing coupled with a large spatial extent of firing fields (current study). Nonetheless, also parietal neurons can code information on the rat's spatial position and may change firing rate during prolonged stretches on a maze⁵²⁻⁵⁴. Thus, these suggested differences await further testing in a direct comparison by simultaneous PPC and PRH recordings during maze behavior."

REVIEWERS' COMMENTS:

Reviewer #1 (Remarks to the Author):

The authors have answered my final query - although it is still not clear to my mind whether the perirhinal firing should be considered spatial (similar to the large firing fields in ventral hippocampus), the data/analyses are now presented for the reader to make up their own mind.

Reviewer#2

(no further comments)

Reply to reviewers' comments on "Firing patterns of perirhinal neurons are sustained across chunks of task environment" by J. J. Bos et al., Revised manuscript for Nature Communications.

REVIEWERS' COMMENTS:

Reviewer #1 (Remarks to the Author):

The authors have answered my final query - although it is still not clear to my mind whether the perirhinal firing should be considered spatial (similar to the large firing fields in ventral hippocampus), the data/analyses are now presented for the reader to make up their own mind.

We thank the reviewer for taking another look at our manuscript. We like the critical stance and believe the manuscript has benefited from his/her comments. We cannot make a direct comparison between the Perirhinal cortex and the Ventral Hippocampus because in addition to the Perirhinal cortex we recorded the Dorsal Hippocampus in the study. We do not mean to claim that the Perirhinal is a spatial analogue of the Ventral Hippocampus, showing large place fields scattered throughout space. However, we feel we present strong evidence that the Perirhinal responses are locked to the spatial layout of our figure-8 maze. These responses appear much more structured in spatial on- and offsets than Ventral Hippocampal place fields as reported by Kjelstrup et al.2008. That said we agree that a direct comparison between Perirhinal responses and the Ventral Hippocampus would be very interesting. Finally, we presented several arguments why the current findings support that perirhinal firing incorporates a spatial dimension – even though more experiments will be needed to decide on the allocentric nature of this spatial element (see also our Discussion section: "It remains to be determined whether the environmental chunks coded by PRH neurons are truly spatial (in the sense of representing positional information allocentrically), and indeed the role of proximal versus distal cues and contextual elements in shaping this code awaits further investigation. Nonetheless, the current results show that task-related variables, such as sensory cue configurations, and correctness of behavioral performance, do not explain Left-Right discriminations").

Reviewer#2

(no further comments)

We would like to thank the reviewer for his/her time and providing critical comments and useful questions.